# Allometric equations and wood density parameters for estimating aboveground and woody debris biomass in Cajander larch (*Larix cajanderi*) forests of Northeast Siberia

Clement J. F. Delcourt[1], Sander Veraverbeke[1]

[1]Faculty of Science, Vrije Universiteit Amsterdam, Amsterdam, 1081 HV, The Netherlands

*Correspondence to*: Clement J. F. Delcourt (c.j.f.delcourt@vu.nl)

**Abstract.** Boreal forests are particularly vulnerable to climate warming which increases the occurrence of natural disturbances, such as fires and insect outbreaks. It is therefore essential to better understand climate-induced changes in boreal vegetation dynamics. This requires accurate estimates of variations in biomass across regions and time. This remains challenging in the extensive larch forests of Northeast Siberia because of the paucity of allometric equations and physical properties of woody debris needed for quantifying aboveground biomass pools from field surveys. Our study is the first to present values of mean squared diameter (MSD) and specific gravity that can be used to calculate fine dead and downed woody debris loads in Cajander larch (*Larix cajanderi*) forests using the line-intersect sampling approach. These values were derived from field measurements collected in 25 forest stands in the Republic of Sakha, Russia, and compared with values reported for other prevalent boreal tree species. We developed allometric equations relating diameter at breast height (DBH at 1.3 m) to stem wood, stem bark, branches, foliage, and aboveground biomass based on measurements of 63 trees retrieved from previous studies. Differences between our allometric models and existing equations were assessed in predicting larch aboveground biomass in 53 forest stands sampled in the Republic of Sakha. We found that using fine woody debris (FWD) parameters from other boreal tree species and allometric equations developed in other regions may result in significantly lower biomass estimates in larch-dominated forests of Northeast Siberia. The FWD parameters and allometric equations presented in our paper can be used to refine estimates of aboveground biomass in Cajander larch forests in Northeast Siberia.

## 1 Introduction

The boreal forest represents the largest remaining forest on Earth and covers large areas of the high northern latitudes (FAO, 2020). The boreal forest stores nearly a quarter of the global terrestrial carbon pool (Scharlemann et al., 2014), and as such plays a major role in the global carbon cycle. In addition to its role of a long-term carbon sink (Goodale et al., 2002; Gurney et al., 2002; Pan et al., 2011), boreal forests influence energy and water fluxes through biogeophysical processes associated with surface albedo and evapotranspiration (Bonan et al., 1992; Thomas and Rowntree, 1992). The accelerated warming observed in the boreal regions (Serreze et al., 2009; Cohen et al., 2014) is predicted to increase the extent, frequency, and severity of natural disturbances such as fires (Kasischke and Turetsky, 2006; Flannigan et al., 2009; De Groot et al., 2013)

and insect outbreaks (Bale et al., 2002; Gustafson et al., 2010), thereby invoking aboveground biomass losses. These changes in land cover may have significant impacts on multiple forcing agents, including from greenhouse gases and aerosols, and through alterations in surface albedo (Bonan et al., 1995; Randerson et al., 2006). Quantifying variations in forest biomass and productivity across regions, ecosystems, and time is therefore essential for assessing climate-induced changes in boreal vegetation dynamics and feedback mechanisms.

Estimating forest carbon stocks is possible from various techniques from local field inventories to spaceborne assessment at regional to global scales (Picard et al., 2012). Remote sensing is an effective tool for forest structure assessments as it provides repeated information over large areas of interest in a cost-effective manner. Indirect measurements of biomass stocks are based on relationships between remotely sensed proxies and detailed field measurements of biomass. In addition to calibrating and validating remotely sensed measurements and constraining biogeochemical models, field measurements

provide a level of detail of the vertical and horizontal structure of biomass that are unmatched from other methods. For example, field-based surveys are required to assess the spatial distribution of dead and downed woody debris (DWD) on the forest floor (Woodall et al., 2008).

DWD is a major component in forest ecosystem functioning. It plays a fundamental role in carbon storage (Woodall et al., 2013), influences other nutrient cycles (Finér et al., 2003; Graham and Cromack, 1982), facilitates tree regeneration (Weaver

et al., 2009), and serves as habitat for several plant and animal species (Harmon et al., 1986; Freedman et al., 1996). DWD is also an important controlling factor of forest fire behavior (Rothermel, 1972). DWD fuel load affects surface fire intensity (Byram, 1959; Alexander, 1982) and can trigger the transition between surface and crown fire regimes (Van Wagner, 1977). Fuel size is also critical in fire behavior prediction as the surface area-to-volume ratio determines fuel moisture and combustion efficiency (Byram, 1959). Ignition and consumption are generally larger for smaller diameter pieces that exhibit

high surface exposure. Quantification of fine and medium-sized dead and downed woody material is therefore essential for assessing fire effects and forest carbon dynamics.

The line-intersect method is one of the most common field-based approaches to sample woody debris in forest stands (Warren and Olsen, 1964; Van Wagner, 1968, 1982; Brown, 1971; Brown and Roussopoulos, 1974; Brown et al., 1982). This technique allows estimation of the total biomass by measuring the diameter or cross-sectional area of all dead and

downed woody pieces at their intersection with a transect line (Warren and Olsen, 1964). As this sampling approach can be tedious and time-consuming for smaller pieces, fine woody debris (FWD, < 7 cm in diameter) are generally tallied by diameter size class using a go/no-go sizing gauge, and load for any size class can be retrieved from the number of intercepts over the transect line (Van Wagner, 1982). One of the advantages of this method is that it significantly reduces sampling time in the field, but locally derived species-specific values for specific gravity, tilt angle and diameter for each size class are

needed. Such data have been published for most of the North American boreal forest tree species (e.g., Brown, 1974; Sackett, 1980; Delisle et al., 1988; Nalder et al., 1999). However, to the best of our knowledge, there is no available data for larch species in boreal forests of Northeast Siberia.

Destructive harvest is the most accurate approach for estimating tree biomass, yet this method requires intensive sampling efforts and complete harvesting of a forest stand is time consuming and not suitable for long-term monitoring. To overcome these limitations, one standard technique is to apply allometric equations to convert easily measured variables, such as tree diameter and height, to biomass for each individual tree in a given area. Allometry examines size-correlated variations in rates of plant growth (Huxley and Teissier, 1936; Niklas, 1994) and can be applied to the different parts of a tree (e.g., stem, foliage, and branches). By sampling individual trees over a size range of interest, allometric equations can be developed using regression models for estimating the relationship between tree component biomass and one or more size parameters.

Numerous allometric equations have been published for North American boreal tree species (Smith and Brand, 1983; Penner et al., 1997; Ter-Mikaelian and Korzukhin, 1997). In contrast, allometric relationships are more limited in the larch forests of Northeast Siberia, which differ from other boreal forests because they consist of deciduous needleleaf trees from the genus *Larix* Mill. Characterized by its adaptability to grow on permafrost terrain, Siberian larch species (*Larix sibirica* Ledeb., *Larix gmelinii* (Rupr) Rupr., *Larix cajanderi* Mayr.) have evolved physiological and morphological features to endure the long and extremely low air and soil temperatures as well as vigorous cryogenic processes in permafrost terrain. These forests account approximately for 20 % of the world's boreal forests (Osawa and Zyryanova, 2010), store vast amount of carbon, mainly in organic soils (Matsuura and Hirobe, 2010), and contribute to the stability of the permafrost in a warming climate (Abaimov, 2010; Stuenzi et al., 2021). Despite their critical importance to the global carbon cycle, larch forest ecosystems, especially the remote Cajander larch (*Larix cajanderi* Mayr.) forests of northeastern Siberia, are understudied and little is known about their response to climate change.

Most of *L. cajanderi* habitat is located eastwards of the Verkhoyansk Mountain Range where very large differences between summer and winter temperatures prevail (Abaimov, 2010; Chevychelov and Bosikov, 2010). *L. cajanderi* can grow on various soil types, from poorly developed stony soils of mountainous terrains to rich alluvial soils, and ranges in size from dwarf and shrub-like forms to 25-m tall trees. It is the least heat-demanding larch species and is well adapted to the underlying continuous permafrost as it can resist freezing temperatures and develop adventitious roots as the active layer deepens (Abaimov, 2010). Few studies have reported allometric equations for *L. cajanderi* in Northeast Siberia (Kajimoto et al., 2006; Alexander et al., 2012), but local allometric equations have been developed based on limited tree harvests near the forest-tundra ecotone (Chersky, 68.74° N, 161.40° E, 100 m above sea level), and the altitudinal treeline (about 100 km west of Oymyakon, 63.45° N, 142.77° E, 1160 m a.s.l). As conifer biomass accumulation and allocation are controlled by climate, soil conditions and stand characteristics (Gower et al., 1995), there may be limitations to applying site-specific allometric equations over large areas away from the measurement sites.

The objective of this study was to expand the biomass allometry of *L. cajanderi* for Northeast Siberia using raw measurements from a comprehensive dataset of biomass measurements in Eurasia (Schepaschenko et al., 2017). We developed allometric equations relating diameter at breast height (DBH) to stem wood, stem bark, foliage, branches, and total aboveground biomass. In addition, we sampled fine woody debris of *L. cajanderi* using the line-intersect method in Siberian larch-dominated forest stands. We derived diameter and specific gravity values per woody debris size-classes. The

allometric equations and wood density parameters presented in this work will be of use to researchers that want to quantify aboveground and woody debris biomass in Cajander larch forests of Northeast Siberia. This data will facilitate improved quantification and understanding of the dynamics of these two major boreal forest carbon pools in Northeast Siberia.

**2 Methods**

**2.1 Fine woody debris sampling**

The line-intersect method is a widely used approach to quantify FWD lying on the ground in a forest stand (Warren and Olsen, 1964; Van Wagner, 1968; Brown, 1971). It requires measuring the diameter of each piece of wood at its intersection with a transect line which can be considered as a strip of infinitesimal width containing a series of cross-sectional areas (Van

Wagner, 1982). The sum of cross-sectional areas divided by the length of the transect line can be converted to volume by multiplying both numerator and denominator by width. FWD load, or weight per unit ground area, is then obtained from Eq. (1) by multiplying the volume by the specific gravity of wood as follows (Van Wagner, 1982):

$$W = \frac{\pi}{2} \times \sum d^2 \times \frac{\pi}{4} \times \frac{G}{L},$$ (1)

where $W$ is the FWD load, $\frac{\pi}{2}$ is a probability factor that allows to sum the cross-sectional areas as circles, $d$ is the piece

diameter, $\frac{\pi}{4}$ is the factor required to convert $d^2$ into circular area, $L$ is the length of the transect line, and $G$ is the specific gravity in units of weight per unit volume. Equation (1) assumes that woody pieces are horizontal and does not account for ground slope. To minimize the bias related to tilted pieces that are less likely to be intercepted by the transect line, $W$ can be multiplied by a correction factor equal to the secant of the piece tilt angle relative to horizontal (Brown and Roussopoulos, 1974). Similarly, a correction factor can be calculated from the ground slope angle as follows (Brown, 1974):

$$s = \sqrt{1 + \tan^2\alpha},$$ (2)

where $s$ is the slope correction factor, $\alpha$ is the ground slope angle (degrees). Consequently,

$$W = \frac{\pi^2 \times G \times \sec h \times \sum d^2 \times s}{8 \times L},$$ (3)

where $h$ is the angle between the piece and the horizontal plane (degrees). Measuring diameter on each intersected piece along a transect line can be tedious and time-consuming, especially where small pieces are abundant. In practice, FWD are

tallied by diameter size class using a go/no-go sizing gauge, and the number of intercepts over the transect line is reported for each class (Brown, 1974; McRae et al., 1979). Therefore, the term $\sum d^2$ in Eq. (3) is replaced by $\sum_i n_i \times D_i^2$, where $n_i$ is the number of intercepts over the transect line within the diameter size class $i$, and $D_i$ is the representative class diameter (Van Wagner, 1982). The quadratic mean diameter (QMD) is generally used as the appropriate class diameter so that load for any species can be calculated as follows (Van Wagner, 1982; Nalder et al., 1999):

$$W_i = \frac{\pi^2 \times G_i \times \sec h_i \times n_i \times QMD_i^2 \times s}{8 \times L},$$ (4)

where $W_i$ is the FWD load (t ha$^{-1}$), $G_i$ is the specific gravity (g cm$^{-3}$), $h_i$ is the piece tilt angle (degrees) of the diameter size class $i$, and QMD$_i$ is the quadratic mean diameter (cm) of the size class $i$ given by:

$$\text{QMD}_i = \sqrt{\frac{\Sigma_i d_i{}^2}{n_i}}. \tag{5}$$

To determine wood density and size parameters for *L. cajanderi*, we sampled natural forest stands within the boreal forests of the Republic of Sakha (Russia) in the summer of 2019 (Fig. 1; Table 1). Our study sites were located on both sides of the Lena River within the Central Yakutian Lowland (0–200 m a.s.l) and the Near-Lena Plateau (200–500 m a.s.l) (Chevychelov and Bosikov, 2010). *Larix cajanderi* largely prevails in these areas of continuous permafrost with inclusion of *Pinus sylvestris* communities (Isaev et al., 2010). We selected larch-dominated stands within a gradient of vegetation structure, stand age, and landscape position. At each site, we established a 30 m × 30 m quadrant, within which measurements were made along a 2 m × 30 m belt transect in the north-south direction intersecting the quadrant's centroid. We recorded latitude, longitude, elevation, slope, aspect, and a general site description. To characterize stand structure and composition, we inventoried every tree within the belt transect by recording species and DBH, and calculated stem density and basal area. We estimated stand age from basal tree disks or increment cores collected on five trees of the dominant cohort.

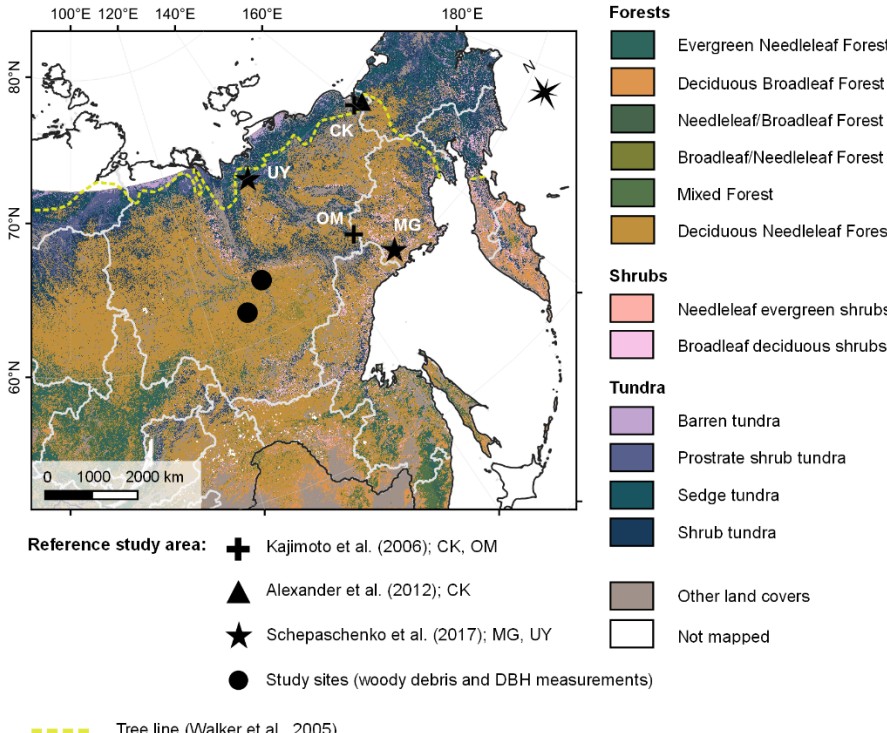

**Figure 1. Field site locations of existing studies on *Larix cajanderi* allometry in Northeast Siberia (Kajimoto et al., 2006; Alexander et al., 2012) and biomass measurements used in this study for the development of new allometric equations.** Locations of the *L. cajanderi* forest stands sampled for fine woody debris analysis and diameter at breast height (DBH) measurements are shown as black circles. Background is the land cover map of Northern Eurasia (Bartalev et al., 2003). CK: Chersky; MG: Magadan Oblast; OM: Oymyakon; UY: Ust-Yansky district.

**Table 1. Stand characteristics of the study sites located in the Republic of Sakha, Russia.** *G*: wood specific gravity; MSD: mean squared diameter; DBH: diameter at breast height; std: standard deviation.

| Forest stand characteristics | Site location | |
|---|---|---|
| | Batamay | Yert |
| Latitude | 63.52° N | 62.02° N |
| Longitude | 129.42° E | 125.79° E |
| Number of stands | 30 | 23 |
| *Number of stands with measurements/estimates for* | | |
| Fine woody debris, *G* and MSD | 4 | 21 |
| Fine woody debris biomass | 25 | 22 |
| DBH, tree biomass | 30 | 23 |
| *Larch trees* | | |
| Mean density ± std (range) (thousand trees ha$^{-1}$) | 10.55 ± 9.93 (1.00–45.33) | 13.50 ± 13.69 (0.17–46.33) |
| Mean tree age ± std (range) (years) | 83.2 ± 40.9 (9–214) | 97.1 ± 39.1 (40–162) |
| Mean DBH ± std (range) (cm) | 5.48 ± 3.56 (1.20–15.08) | 5.09 ± 5.14 (0.22–18.56) |

FWD was sampled for *L. cajanderi* using the line-intersect approach. In each site, we inventoried every piece (e.g., twigs, limbs, branches) that intersects the 30 m transect line (west side of our belt transect) in each of the following five roundwood
diameter size classes: 0–0.49 cm, 0.50–0.99 cm, 1–2.99 cm, 3–4.99 cm, and 5–6.99 cm (McRae et al., 1979), hereafter referred to as classes I to V. For the first five pieces in each size class, we measured the distance along the transect (m) and their diameters (cm) at their points of intersection with the transect line (Fig. 2) using a caliper to 1 mm. These pieces were systematically collected and returned to the laboratory for specific gravity determination using the water displacement method in accordance with the ASTM International D2395–14 Method B-II (ASTM International, 2014). First, we measured
the oven-dry mass of each sample by drying to constant mass (i.e., less than 0.2 % mass change over a five-hour period of drying) at 105 °C (Roussopoulos and Johnson, 1973). After drying, the water displacement method requires to seal the surface of woody pieces to prevent bias in volume determination resulting from water absorption during immersion. Each oven-dried sample was then covered with a thin impermeable layer by immersion in hot liquid paraffin (solidification point 57–60 °C, 0.90 g cm$^{-3}$ at 20 °C), and the mass of the coated piece was measured again before volume determination. As
suggested by Nalder et al. (1999), we used a paraffin temperature of 180 °C to minimize the error associated with the use of a finite thickness paraffin layer. Using a pointed, slender rod, each sample was submerged in a container of water placed on a

balance. The mass of water after immersion in grams corresponds to the volume of the immersed sample in cubic centimeters. Finally, specific gravity of each sample was determined as follows (ASTM International, 2014):

$$G = \frac{K \times m_0}{V_0},$$ (6)

where $G$ is the specific gravity (g cm$^{-3}$), $K$ is a constant equal to 1 when the mass is in grams and the volume is in cubic centimeters, $m_0$ is the oven-dry mass (g), and $V_0$ is the oven-dry volume (cm$^3$) given by:

$$V_0 = m_{w,disp} - \frac{m_{coated} - m_0}{\rho_{paraffin}},$$ (7)

where $m_{w,disp}$ is the mass of water displaced (g), $m_{coated}$ is the mass of the oven-dried sample after immersion in the paraffin (g), and $\rho_{paraffin}$ is the density of the paraffin wax (g cm$^{-3}$).

For each site and size class, we calculated the mean specific gravity and mean squared diameter (MSD) using a similar approach to Sackett (1980) and Nalder et al. (1999). Defined as the sum of the squared diameters divided by the number of samples, MSD is a direct measure of cross-sectional area. It is therefore more adequate than QMD to interpret differences in dead and downed woody debris biomass between species and regions (Nalder et al., 1999). All statistical analyses were performed using R statistical software version 4.0.3 (R Core Development Team, 2021). We compared specific gravity and

MSD as a function of diameter size class using a one-way analysis of variance (ANOVA) followed by a Tukey-Kramer post hoc analysis for multiple comparisons. Each MSD value was divided by the square of the arithmetic class center (ACC) of the corresponding size class (i.e., 0.25, 0.75, 2, 4, and 6 for classes I to V) to test for the effect of size classes relative to class midpoint. In cases where the assumptions of ANOVA were not met, we used the non-parametric Kruskal-Wallis test and the Wilcoxon rank sum test for pairwise comparisons with Benjamini-Hochberg corrections for multiple testing as implemented

in the R 'stats' package.

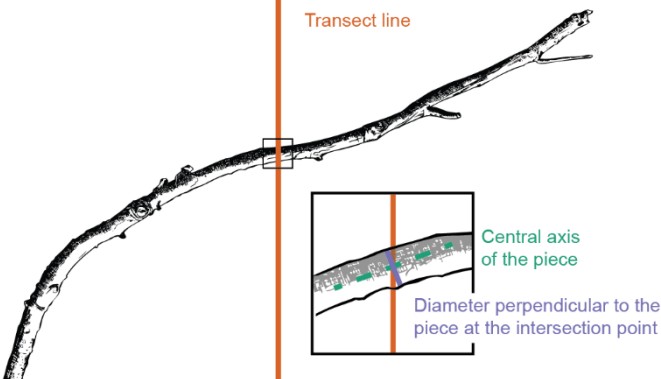

**Figure 2. Fine woody debris (FWD) diameter measurement using the line-intersect method.** Diameter was measured perpendicular to the central axis of the sampled piece at the intersection point with the transect line.

To simplify FWD loads calculations, size-class specific values of specific gravity, MSD, and tilt angle can be combined into

a single factor $M$ (g cm$^{-1}$) as follows (Nalder et al., 1999):

$$M_i = \frac{\pi^2 \times G_i \times \sec h_i \times MSD_i}{8}.$$ (8)

While tilt bias can be significant in fresh logging slash where smaller pieces are attached to larger ones (e.g., correction factor as high as 1.38) (Brown and Roussopoulos, 1974), a FWD inventory made in harvested and natural Canadian boreal forest sites indicated that it was generally not large (Nalder et al., 1999). To facilitate comparisons with other species and regions, we used a tilt correction factor of 1.13 as suggested by Brown (1974). The single factor $M$ defined in Eq. (8) represents the load per intercept per meter of transect and can then be used to calculate FWD loads for each diameter size class $i$ using the formula:

$$W_i = \frac{n_i \times M_i}{L}.$$ (9)

We used Eq. (9) to calculate FWD loads at 47 of 53 forest stands where *Larix cajanderi* fine woody debris were observed (Table 1) using $M$ factors derived from this study. For each site, we derived the percentage difference between these estimates and FWD loads computed using $M$ factors from other species and boreal regions.

## 2.2 Biomass data

We retrieved raw biomass data from a comprehensive dataset of destructive sampling measurements compiled by Schepaschenko et al. (2017). This dataset summarizes measurements from about 1200 experiments conducted between 1930 and 2014 in Eurasia. This dataset provides valuable data that were previously difficult to access for the non-Russian scientific community due to limited publications in English language literature. For each sample plot, the dataset consists of forest stand parameters (e.g., tree species composition, stand age, stem density) as well as tree level variables including DBH, stem height, age, and biomass. Tree biomass is reported in kilograms of dry matter for four components, including stem, stem bark, branches, and foliage, as well as for the total aboveground pool. All the data contained in this dataset were collected using similar sampling strategies. We selected raw harvest data from *L. cajanderi* stands sampled as part of two experiments conducted in the Magadan Oblast and the Ust-Yansky district (Republic of Sakha), Russia (Schepaschenko et al., 2017) (Fig. 1). Table 2 summarizes the characteristics of larch-dominated forest stands sampled in both sites.

**Table 2. Summary of tree samples harvested in the Magadan Oblast and the Ust-Yansky district, Russia (Schepaschenko et al., 2017).** The ranges in stand density, tree age, diameter at breast height (DBH), and tree height ($H$) are provided for each experiment.

| Location | Latitude | Longitude | Number of stands | Range Density (trees ha$^{-1}$) | Number of trees | Range Tree age (years) | DBH (cm) | $H$ (m) |
|---|---|---|---|---|---|---|---|---|
| Magadan Oblast | 60.50 | 148.00 | 7 | 250–860 | 43 | 30–424 | 3.9–52.8 | 1.9–30.0 |
| Ust-Yansky district | 69.88 | 135.59 | 2 | 600–1521 | 20 | 57–154 | 1.8–18.9 | 2.2–11.8 |

## 2.3 Development of allometric equations

We developed site-specific allometric equations relating tree DBH to component biomass and total aboveground biomass by fitting power functions of the form:

$$Y = a \times \text{DBH}^b ,\tag{10}$$

where $Y$ is the dry biomass (kg), DBH is the tree diameter at breast height (cm), and $a$ and $b$ are the regression coefficients. Power relationships are commonly used when describing plant allometry and can be approximated following different regression techniques. We first applied a double logarithmic transformation of both biomass and diameter variables as in Eq. (11) and fitted the linear regression using the ordinary least squares method.

$$\ln Y = \ln a + b \times \ln \text{DBH} \tag{11}$$

The logarithmic transformation introduces a downward bias in the prediction of plant biomass (Baskerville, 1972). A correction factor ($CF$) was therefore calculated using Eq. (12) for each regression and multiplied to the value of $a$ to correct for this systematic bias (Sprugel, 1983):

$$CF = \exp\left(\frac{\hat{\sigma}^2}{2}\right) ,\tag{12}$$

where $\hat{\sigma}$ is the residual standard deviation. The log-log form depicted in Eq. (11) has been widely used to model allometric relationships as it corrects for heteroscedasticity and facilitates statistical comparison between several equations compared to curvilinear regressions (Gower et al., 1999). However, the performance of the double logarithmic transformation model may be reduced when predicting biomass over a wide range of tree diameters (Bond-Lamberty et al., 2002).

Equation (10) was then fitted using the weighted nonlinear regression model as implemented by the function 'nls' in the R 'stats' package. Originally developed in forestry in the 1960s (Cunia, 1964; Wharton and Cunia, 1987), the weighted least squares method can be used to treat models which have non-random residuals (Brown et al., 1989; Parresol, 1999; Moore, 2010). This method involves assigning each sample a positive weight $w_j$ as follows (after Picard et al., 2012):

$$\varepsilon_j \sim \mathcal{N}\left(0, \sigma_j\right) ,\tag{13}$$

$$w_j \propto \frac{1}{\sigma_j^2} ,\tag{14}$$

where $\varepsilon_j$ is the residual error and $\sigma_j$ is the residual standard deviation for each observation $j$. In biomass studies, the heteroscedasticity of the residuals is often approximated by a power relationship between the residual variance and the diameter of trees (Picard et al., 2012) such that:

$$\sigma_j = k \times \text{DBH}_j^{\,c} ,\tag{15}$$

where $k > 0$ and $c \geq 0$. Consequently,

$$w_j \propto \text{DBH}_j^{-2c} .\tag{16}$$

As shown in Eq. (16), the weighting is defined by the value of the exponent $c$ that must be fixed prior to fitting the nonlinear regression. For each site and biomass component, the value of $c$ was determined by (1) approximation of the conditional variance of biomass, and (2) visual assessment of the plots of the weighted residuals against the fitted values (Table B1; Fig.

B1; Fig. B2). In the first approach, we followed Picard et al. (2012) by dividing DBH values in $K$ classes centered on $DBH_k$, where $k = \{1,\ldots,K\}$. We selected $K = 5$ in this study and visually checked that the power model was appropriate for modeling the residual variance. A linear regression was fitted between the standard deviation of biomass and the median DBH of each class $k$ using a double logarithmic transformation and the value of the exponent $c$ was approximated as the slope of this regression. We report regression coefficients and root mean square errors (RMSE) for each linear and nonlinear regression equation. Standard errors of the regression coefficients are also provided for uncertainty propagation purposes. To determine whether there were differences in allometric equations among sites, we fitted one model where regression parameters varied from site to site (reduced model) and another model where the parameters were identical for both sites (full model). We tested the null hypothesis that $a$ and $b$ parameters did not differ between the two sites using the extra sum-of-squares $F$-test (Motulsky and Christopoulos, 2004). Significant test results would suggest site-specific variations in the allometric relationships between DBH and component biomass. Our allometric models were compared with existing biomass equations for *L. cajanderi* published by Kajimoto et al. (2006) and Alexander et al. (2012). We assessed differences between models in predicting larch aboveground biomass in 53 forest stands located in the study area described in Sect. 2.1 (Table 1). In each stand, we inventoried every tree within the belt transect and measured DBH (at the height of 1.3 m from the base). Significant differences among models were assessed using the non-parametric Kruskal-Wallis test and the Wilcoxon rank sum test for pairwise comparisons (0.05 significance level).

## 3. Results and discussion

### 3.1 Fine woody debris

We sampled a total of 25 forest stands, measuring diameter on 223 pieces and specific gravity on 95 pieces of dead and downed wood of *L. cajanderi*. Summary values of MSD and specific gravity are reported in Table 3. Thirteen sites were intermediate aged (58.1 ± 6.8 years old) and 11 were mature (123.5 ± 24.5 years old). Basal areas and tree densities ranged from 1.9 to 15.7 $m^2$ $ha^{-1}$ and 1170 to 69830 stems $ha^{-1}$, respectively. The average dominance of *L. cajanderi* was 64 %, and larch trees co-occurred with either Scots pine (*Pinus sylvestris*) or silver birch (*Betula pendula*) or both in 21 of the 25 sampled sites. Specific gravity ranged from 0.625 to 0.672 g $cm^{-3}$ (Table 3) and differed as a function of diameter size classes (ANOVA, $F_{3,91} = 3.22$, $p = 0.026$), with the Class II being significantly higher than Class III (Tukey, $p < 0.05$). Site-level MSD/$ACC^2$ values were normally distributed (Shapiro-Wilk, $p = 0.08$) but the assumption of homogeneity of variance was violated (Levene, $p < 0.05$). The log-transformed data did not satisfy the assumptions of normality and homoscedasticity (Shapiro-Wilk, $p < 0.05$; Levene, $p < 0.05$). MSD varied significantly across size classes (Kruskal-Wallis, $\chi^2 = 21.8$, $df = 4$, $p < 0.001$) with Class II being significantly different from Class III (Wilcoxon, $p < 0.05$) and Class IV (Wilcoxon, $p < 0.001$).

**Table 3. Mean squared diameter (MSD) and mean specific gravity (G) of fine dead and downed woody debris by diameter size class.** The number of pieces for each size class is reported in brackets. Letters represent significant differences ($p < 0.05$) between size classes determined using the Tukey-Kramer test (G) or the Wilcoxon rank sum test (MSD) for multiple comparisons.

| Diameter size class | MSD ($cm^2$) | $G$ ($g\ cm^{-3}$) |
|---|---|---|
| *Larix cajanderi* | | |
| I (< 0.5 cm) | 0.04 (6) [ab] | Not measured |
| II (0.5–0.99 cm) | 0.67 (47) [a] | 0.672 ± 0.049 (30) [a] |
| III (1.0–2.99 cm) | 3.27 (123) [b] | 0.633 ± 0.066 (40) [b] |
| IV (3.0–4.99 cm) | 13.89 (32) [b] | 0.625 ± 0.052 (13) [ab] |
| V (5–6.99 cm) | 35.16 (15) [ab] | 0.659 ± 0.069 (12) [ab] |

Specific gravity was the highest in Class II and then decreased with larger pieces until Class V where it had intermediate values (Table 3). This pattern in the variation of $G$ across diameter size classes is similar to that of *Larix laricina* (tamarack) in the boreal forests of western Canada (Nalder et al., 1999), although our values for *L. cajanderi* are higher. Discrepancy in specific gravity between species can be attributed to differences in live tree stem wood specific gravity, but also to decomposition rates and growth ring patterns that are both influenced by climate. In particular, the tighter the growth rings, the greater the proportion of cell walls in the wood cells. It is worth noting that our approach to measure specific gravity has some limitations associated with (1) the removal of rotten pieces for which we were not able to determine the volume; (2) the inevitable uptake of moisture between oven-drying and volume determination; and (3) the use of paraffin wax to seal the surface of woody pieces before their immersion in water. The latter may lead to an overestimation of the oven-dry volume resulting from the filling of small cracks with paraffin (Nalder et al., 1999). Furthermore, the number of pieces sampled for specific gravity and MSD determination is relatively small for larger size classes.

Despite these limitations, this study is the first to report measurements of specific gravity and mean squared diameter for *Larix cajanderi* in the boreal forests of Northeast Siberia. Nalder et al. (1999) showed that these two variables can vary substantially between species and regions, emphasizing the need for locally derived species-specific data. When calculated for size classes II, III, IV, and V from values reported in Table 3, the single factor $M$ varies considerably from values derived by Nalder et al. (1999) for common boreal tree species including *Picea mariana* (black spruce), *Picea glauca* (white spruce), and *Larix laricina* in the Canadian Northwest Territories and Saskatchewan (Table 4). For 19 of the 20 comparisons, $M$ was larger for *Larix cajanderi* than for the other species, with a mean difference of 23 % (range: -2–46 %) (Table 4). Fine woody debris biomass averaged 2.19 t ha$^{-1}$ in the 47 forest stands sampled in the Republic of Sakha when calculated from size-class specific $M$ values derived in similar Cajander larch forest ecosystems (Table A1). Figure 3 reveals that using $M$ factors derived for other boreal tree species and regions resulted in lower FWD loads between on average 9 % (*Picea mariana*, Northwest Territories) and 28 % (*Larix laricina*, Saskatchewan). This suggests that using specific gravity and MSD values from other species and/or regions would significantly underestimate dead and downed wood biomass in larch-dominated boreal forests of Northeast Siberia.

**Table 4. Values of the multiplication factor *M* from this study and those from other tree species in the Canadian Northwest Territories and Saskatchewan by diameter size class.** *M* values for *Larix laricina* (tamarack), *Picea glauca* (white spruce), and *Picea mariana* (black spruce) were derived from Eq. (8) using specific gravity (*G*) and mean squared diameter (MSD) values from Nalder et al. (1999). To facilitate comparisons with *M* values from our study, we used a tilt correction factor (sec h) of 1.13 as suggested by Brown (1974).

| Location | Species | Value of *M* per diameter size class (II–V) | | | |
|---|---|---|---|---|---|
| | | II | III | IV | V |
| Northeast Siberia | *Larix cajanderi* | 0.628 | 2.89 | 12.1 | 32.3 |
| Northwest Territories | *Picea glauca* | 0.389 | 2.45 | 10.6 | 22.9 |
| | *Picea mariana* | 0.424 | 2.94 | 11.5 | 25.2 |
| Saskatchewan | *Larix laricina* | 0.338 | 1.85 | 9.63 | 31.1 |
| | *Picea glauca* | 0.397 | 2.47 | 10.1 | 19.5 |
| | *Picea mariana* | 0.380 | 2.48 | 10.7 | 23.1 |

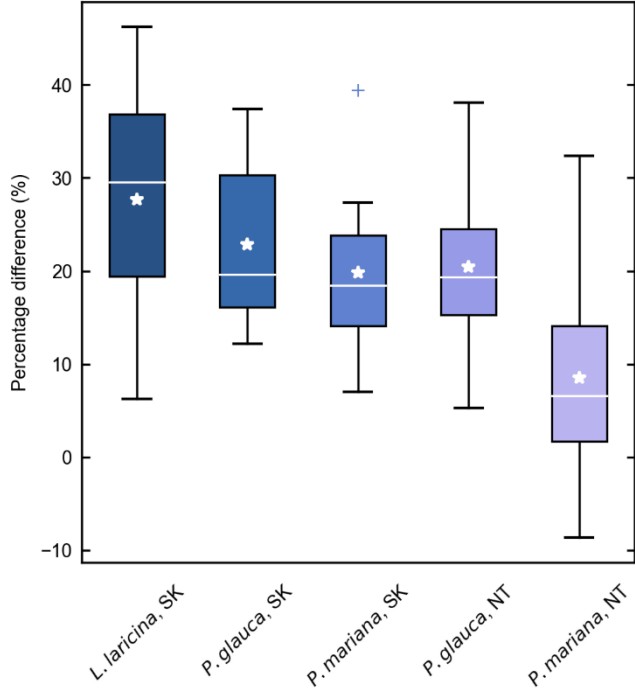

**Figure 3. Percentage difference in fine woody debris (FWD) biomass estimates in 47 larch forest stands (*Larix cajanderi*) near Yakutsk using *M* factors derived for other species and boreal regions.** Differences were calculated from the estimates based on the *M* values developed in this study, such that a positive percentage difference reflects a lower biomass estimate. Each box ranges from the first quartile (Q1) to the third quartile (Q3), with the median and mean indicated by a white horizontal line and star. The whiskers extend from Q1 and Q3 to the minimum and maximum defined as Q1−1.5×IQR and Q3+1.5×IQR respectively, where IQR is the interquartile range (Q3−Q1). Outliers above the maximum or below the minimum are indicated by crosses.

## 3.2 Allometric relationships

In line with previous studies in Northeast Siberia (Kajimoto et al., 2006; Alexander et al., 2012), DBH-based site-specific allometry is a reliable approach for estimating aboveground tree biomass in mature stands of *L. cajanderi*. All equations fitted using linear regressions applied to log-transformed data were statistically significant ($p < 0.001$) (Table C1). At Magadan and Ust-Yansky sites, coefficients of determination ($R^2$) for individual biomass components ranged from 0.84 to 0.96 and 0.93 to 0.99, respectively. Weighted nonlinear regression models resulted in better fits in all cases (Table 5; Fig.

C1). DBH was a significant predictor ($p < 0.05$) of tree components biomass at Magadan (RMSE = 2.60–46.29 kg) and Ust-Yansky (RMSE = 0.51–5.57 kg) sites (Table 5). Higher RMSE observed at Magadan sites are explained by the larger range of DBHs. Most biomass equations for boreal tree species were derived from linear regressions on log-transformed data (Ter-Mikaelian and Korzukhin, 1997; Bond-Lamberty et al., 2002). However, the slopes and intercepts of the relationships between tree diameter and biomass may differ with stand age (Lieffers and Campbell, 1984; Campbell et al., 1985), which

limits the efficiency of the logarithmic transformation for describing plant allometry over a wide range of tree sizes (Bond-Lamberty et al., 2002; Kajimoto et al., 2006). Similar to a previous work on shrub allometry in boreal forests (Berner et al., 2015), our results show that weighted nonlinear regression models can be used to predict plant biomass. Contrary to previous works on Cajander larch allometry, the set of equations presented in this study includes bark as a separate biomass component. Quantifying bark biomass in Siberian larch forests is important, for example when estimating fuel consumption

from wildfires, as bark and wood are characterized by different combustion processes (Lestander et al., 2012).

**Table 5. Allometric equations relating diameter at breast height (DBH) to stem, stem wood, stem bark, branches, foliage, and total aboveground biomass at *Larix cajanderi* sites in the Magadan Oblast and the Ust-Yansky district (Republic of Sakha).** Allometry is expressed as a power-form equation $Y = a \times DBH^b$ where units of $Y$ are in kilograms of dry weight and DBH is in centimeters. The coefficients $a$ and $b$ were derived using weighted nonlinear least squares regressions in which residuals were weighted by $DBH^{-2c}$ to

correct for non-constant residuals. The values of the exponent $c$ for each location and biomass component are reported in Table B1. For each allometry model, the range of DBH used in regression is provided, as well as the number of trees, the standard errors (SE) of the coefficients and the root mean square error (RMSE).

| Location | Component $Y$ (kg) | DBH range (cm) | Number trees | $a$ | $a$ [SE] | $b$ | $b$ [SE] | RMSE (kg) |
|---|---|---|---|---|---|---|---|---|
| *Magadan Oblast* | Stem | 3.9–52.8 | 43 | 0.066 | 0.014 | 2.493 | 0.068 | 40.934 |
| | Stem wood | 3.9–52.8 | 32 | 0.058 | 0.014 | 2.505 | 0.075 | 46.293 |
| | Stem bark | 3.9–52.8 | 32 | 0.006 | 0.002 | 2.556 | 0.110 | 10.509 |
| | Branches | 3.9–52.8 | 43 | 0.031 | 0.013 | 2.136 | 0.144 | 17.082 |
| | Foliage | 3.9–52.8 | 43 | 0.016 | 0.006 | 1.835 | 0.130 | 2.601 |
| | Aboveground | 3.9–52.8 | 43 | 0.097 | 0.019 | 2.423 | 0.062 | 45.071 |
| *Ust-Yansky district* | Stem | 1.8–18.9 | 20 | 0.128 | 0.013 | 2.227 | 0.059 | 5.568 |
| | Stem wood | 1.8–18.9 | 20 | 0.084 | 0.011 | 2.308 | 0.073 | 5.338 |
| | Stem bark | 1.8–18.9 | 20 | 0.046 | 0.007 | 2.006 | 0.078 | 0.816 |
| | Branches | 1.8–18.9 | 20 | 0.105 | 0.026 | 1.757 | 0.115 | 1.536 |
| | Foliage | 1.8–18.9 | 20 | 0.043 | 0.010 | 1.642 | 0.111 | 0.510 |
| | Aboveground | 1.8–18.9 | 20 | 0.260 | 0.039 | 2.062 | 0.074 | 5.285 |

Our regression models performed poorer in predicting foliage and branches biomass, particularly in the Magadan site ($R^2 <$ 0.90), as shown by the higher errors of the regression coefficients (Table 5). This is in agreement with other studies on shrubs (Buech and Rugg, 1995; Berner et al., 2015) and trees (Gower et al., 1997; Bond-Lamberty et al., 2002) that found weaker allometric models for components related to annual production such as foliage and new growth. It is also clear from Fig. 4b, c and Fig. C2b, c that there is more discrepancy between allometric models for these biomass components, which is likely to be caused by local variations in light, water, and nutrient availability (Bond-Lamberty et al., 2002). While some studies have used crown-base diameter and sapwood area to improve biomass estimation for foliage and branches (Waring et al., 1980; O'Hara, 1988; Comeau and Kimmins, 1989; Osawa, 1990), Kajimoto et al. (2006) found that regression models based on these variables did not always result in better fits in *L. gmelinii* and *L. cajanderi* stands. Including tree height in addition to DBH as independent variables is another alternative often used for boreal tree species (Schmitt and Grigal, 1981; Ker, 1984; Harding and Grigal, 1985), including Siberian larch species (Sawamoto et al., 2003). However, it may be more difficult to measure tree height accurately in the field and more time-consuming to apply such an allometric model for biomass estimation, especially in dense forest stands. We tested tree height ($H$) as a predictor of tree components biomass (Table C2) and found that the improvement in model fits may not be large enough to counterbalance the additional practical difficulties associated with making in situ tree height measurements.

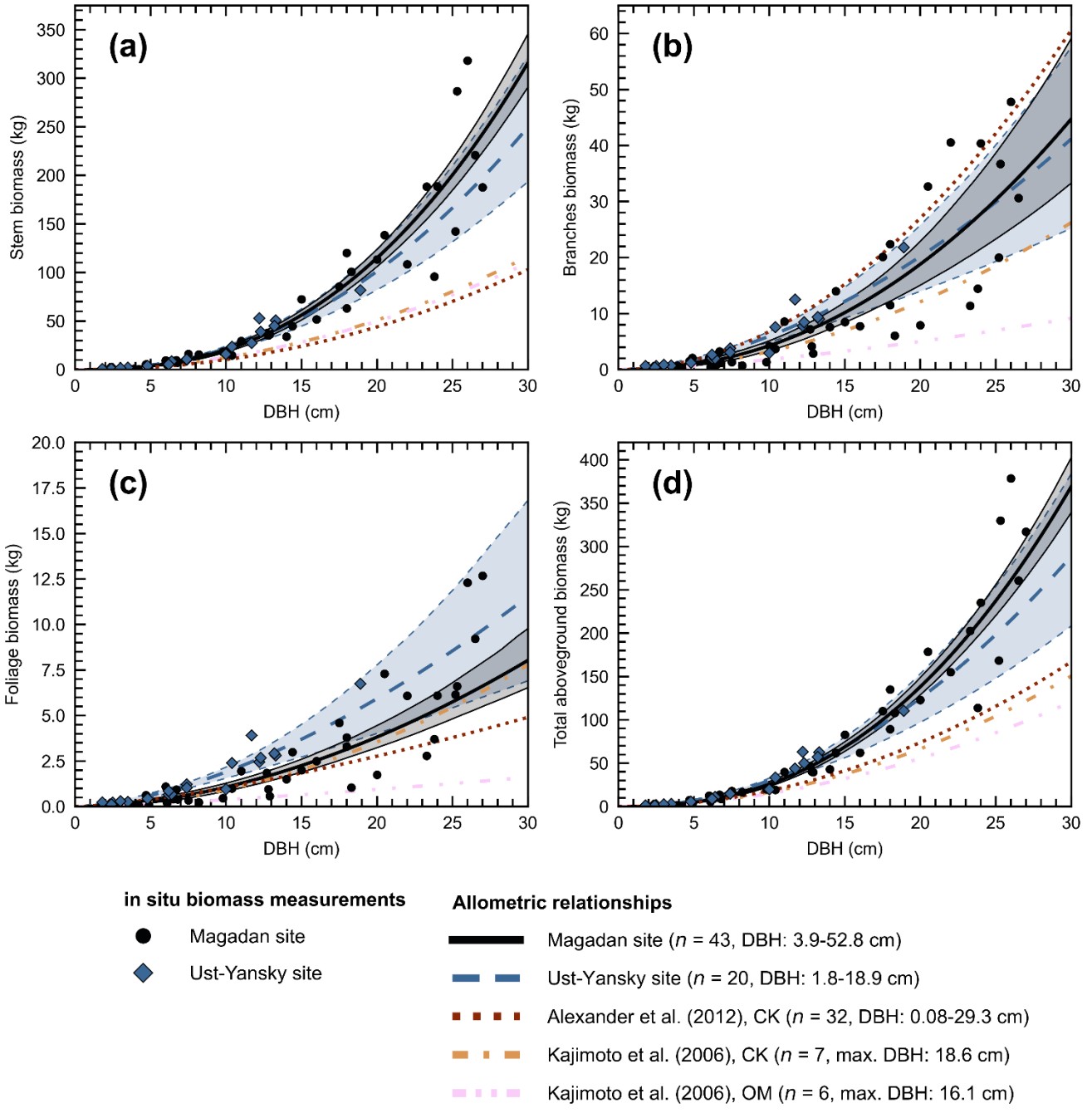

**Figure 4. Allometry models developed in northeastern Siberia for *Larix cajanderi* relating diameter at breast height (DBH) to (a) stem, (b) branches, (c) foliage, and (d) aboveground tree biomass.** For each model, the number of trees harvested (*n*) and the range or maximum (max.) value of DBH are provided. Regression coefficients are given in Table 5 for allometric equations derived at Magadan and Ust-Yansky sites. For allometric relationships developed in Chersky (CK) and Oymyakon (OM) areas, regression coefficients can be found in Kajimoto et al. (2006) and Alexander et al. (2012).

Figure 4d shows differences between our newly developed allometric models and existing aboveground biomass equations developed for *Larix cajanderi* trees in Northeast Siberia (Kajimoto et al., 2006; Alexander et al., 2012). To illustrate these differences, we calculated aboveground biomass at 53 forest stands near Yakutsk (Fig. 1; Table 1) using each allometric model. Total aboveground biomass estimated by applying site-specific allometry from the Magadan Oblast averaged at 6.58 $\pm$ 3.94 kg m$^{-2}$ (range: 0–17.9 kg m$^{-2}$), and at 7.21 $\pm$ 3.42 kg m$^{-2}$ (range: 0–15.1 kg m$^{-2}$) when using the equation developed for

the Ust-Yansky district (Fig. 5). Estimates from allometric relationships developed for the Chersky and Oymyakon areas resulted in significant lower biomass estimates (Wilcoxon, $p < 0.05$), ranging from 3.64 kg m$^{-2}$ to 4.40 kg m$^{-2}$ (Fig. 5). A similar trend was found when predicting stem biomass (Fig. 4a; Fig. C2a). Aboveground biomass is assumed to vary along a latitudinal gradient within larch forests of Northeast Siberia (Usoltsev et al., 2002), ranging from 0.1 kg m$^{-2}$ in northern regions to 18 kg m$^{-2}$ in more productive southern stands (Usoltsev, 2001; Kajimoto et al., 2010). Using the allometric

equations developed in this study, mean aboveground biomass estimates across our larch forest stands were 33–48 % higher than predictions from existing biomass equations. Our estimates were in good agreement with allometry-derived values reported for similar larch forest ecosystems located in nearby areas (Fig. D1). Siewert et al. (2015) reported a mean aboveground tree biomass of 7.2 kg m$^{-2}$ in 12 *L. cajanderi* dominated forest stands in the Spasskaya Pad/Neleger study area (62.23° N, 129.62° E). Our study area is in the contact zone between *Larix cajanderi* and another closely related Siberian

larch species, *Larix gmelinii* (Rupr.) Rupr. (Abaimov, 2010). Schulze et al. (1995) and Sawamoto et al. (2003) used allometric relationships to compute aboveground biomass in *L. gmelinii* forest stands. Their estimates ranged from 4.4 kg m$^{-2}$ (49 years old) to 12.0 kg m$^{-2}$ (125 years old), and 2.0 kg m$^{-2}$ (25 years old) to 10.5 kg m$^{-2}$ (170 years old). The biomass estimates at our study sites are of the same order as those of previous studies in similar mature larch forest ecosystems. Further efforts are needed to verify and validate the utility of our newly developed allometric equations. Part of this effort

could focus on cross-comparing biomass estimates derived from allometric equations with remote sensing measurements from optical and light detection and ranging (LiDAR) sensors (Miesner et al., 2022).

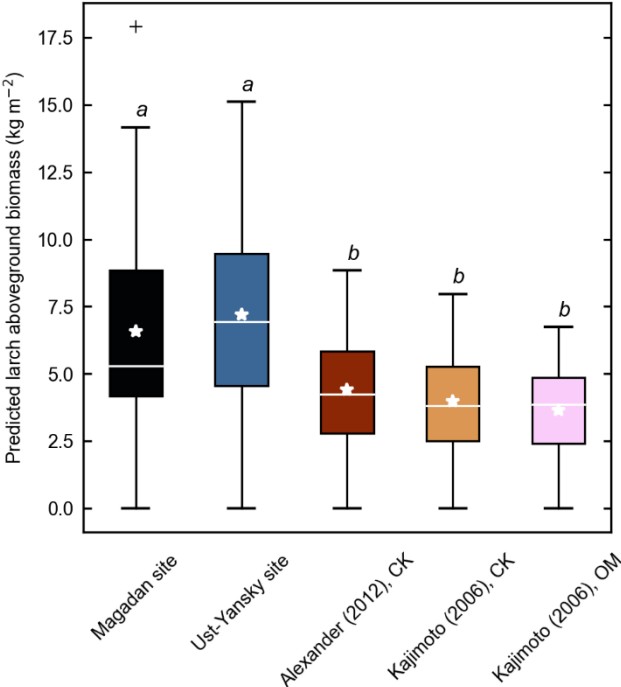

**Figure 5. Larch aboveground biomass predicted for 53 forest stands near Yakutsk using allometry models developed in the Magadan Oblast, the Ust-Yansky district, Chersky (CK) area (Kajimoto et al., 2006; Alexander et al., 2012) and Oymyakon (OM) area (Kajimoto et al., 2006).** Letters represent significant differences ($p < 0.05$) between allometry models determined using the Wilcoxon rank sum test for pairwise comparisons.

Previous studies have shown that caution is needed when applying regional biomass equations in other regions (Bond-Lamberty et al., 2002; Jenkins et al., 2003; Berner et al., 2015). Differences in the relationship between tree diameter and biomass may relate to differences in local site conditions, including stand age, tree density, growing season length, drainage, and nutrient availability (Schulze et al., 1995; Kajimoto et al., 2010). By influencing these factors, permafrost and fire disturbances may shift biomass allocation of Siberian larch trees (Kajimoto et al., 1999; Alexander et al., 2012). Stands structure and dynamics in the Chersky and Oymyakon areas differ with our study sites in the Magadan Oblast and the Ust-Yansky district (Table 2). Allometric equations provided in Alexander et al. (2012) were derived from trees harvested near Chersky in one young stand (15 years old) and two intermediate-aged stands (60 years old), as well as from a mature stand (155 years old) described in Kajimoto et al. (2006). The mature stand (140 years-old) sampled near Oymyakon was located near the altitudinal treeline of *L. cajanderi* on the middle part of a north-facing slope (10°) and consisted of relatively small trees (mean DBH = 5.93 cm, mean *H* = 4.31 m) (Kajimoto et al., 2006). All sites described in this study are underlain by continuous permafrost, however differences in active layer dynamics may contribute to differences in allocation patterns. Soil water and nitrogen (N) availability increase as the active layer deepens, which may allow trees to store more aboveground carbon. In contrast, a shift in proportion of carbon investment from the growth of stem and/or branches to roots

development has been reported in northeastern Siberian larch forests growing on permafrost soils with thin active layers (Kajimoto et al., 1999; Kajimoto et al., 2006). This root-oriented biomass allocation pattern is primarily explained as a response of larch trees to N-poor soils (Kajimoto et al., 2010; Osawa et al., 2010). The particular soil conditions that prevail on north-facing slopes of mountainous terrain, such as lower soil temperature and shallower active layers, as well as the limited direct solar radiation may reduce the rate of photosynthesis and respiration of larch trees (Koike et al., 2010). These limited resources of nutrients and light may explain why larch trees at Oymyakon sites (OM) tended to allocate less aboveground biomass per unit DBH (Fig. 4). Our study shows that using our newly developed allometric equations may result in higher biomass estimates compared to values derived from existing DBH-based biomass equations. This may have important implications for understanding changes in boreal vegetation dynamics, carbon, and energy budgets in a warming climate. Our results suggest that site-specific allometric equations should be applied near the sampling sites from which they were developed. Given the vastness of the area covered by Cajander larch and the comparatively limited number of available allometric equations, a practical challenge arises when one would want to measure tree biomass in areas that are farther away from the four sampling locations described in this study (i.e., the Ust-Yansky district, Chersky and Oymyakon areas in the Republic of Sakha, and the Magadan Oblast). In this case, we recommend using site-specific equations when site parameters (e.g., stand age, DBH, tree height, etc.) closely match with one of the sampling sites. When this is not the case, we recommend that one calculates tree biomass from the five different equations (Fig. 4). This will result in a mean estimate and an uncertainty range. We acknowledge that the number of samples resulting in our newly developed allometric equations is still relatively small, and these equations can further be refined in the future when more data becomes available that covers a wider range of climate conditions, tree ages, and stem sizes. Efforts in this direction are already underway (Miesner et al., 2022).

**Conclusions**

Quantifying regional and temporal variations in biomass of boreal forests is essential for assessing the effects of climate warming on these ecosystems. This study contributes much needed data and equations for estimating aboveground biomass in larch-dominated forests of Northeast Siberia. We collected measurements of specific gravity and mean squared diameter from fine dead and downed woody debris in the Republic of Sakha (Russia). These values, the first available for *Larix cajanderi*, are required to calculate woody debris biomass using the line intersect method. We showed that using data from other dominant boreal tree species may result in large biases.

Allometric equations relating DBH to aboveground tree biomass, as well as individual tree compartments (i.e., stem wood, stem bark, branches, foliage), were developed for two sites in the Ust-Yansky district and the Magadan Oblast (Russia). In agreement with previous studies, our analysis revealed that DBH-based allometry was a simple and reliable approach for estimating *L. cajanderi* biomass in Siberian forests. However, we found significant differences between our newly developed allometric relationships and existing equations, which are likely due to differences in stand characteristics and environmental

conditions at sampled sites. Our results suggest that site-specific allometric equations should be applied with caution in other regions. While our newly developed allometric equations have further extended the geographical coverage of available allometric equations within the range of the *L. cajanderi* forests in Northeast Siberia, our study also demonstrated the critical need for additional field surveys to reduce uncertainties in biomass pools estimates. The data and equations presented in our study contribute to the quantification of aboveground boreal biomass pools of the larch forests of Northeast Siberia.


**Appendices**

 **Appendix A**

**Table A1. Estimates of fine woody debris (FWD) biomass per diameter size class in 47 larch forest stands (*Larix cajanderi*) near Yakutsk (Republic of Sakha, Russia) calculated using *M* factors from Table 4.** Estimates are provided as mean ± standard deviation. For each diameter size class (II–V), the average number of pieces tallied along the transect line in the 47 stands is shown, as well as the range in brackets.

| Location | Species | FWD biomass (t ha$^{-1}$) | | | | |
|---|---|---|---|---|---|---|
| | | II | III | IV | V | all classes |
| | | 6.7 (0–36) | 9.7 (0–44) | 1.0 (0–9) | 0.7 (0–4) | 18.0 (1–82) |
| Northeast Siberia | *Larix cajanderi* | 0.14 ± 0.18 | 0.93 ± 1.07 | 0.41 ± 0.71 | 0.71 ± 1.17 | 2.19 ± 2.07 |
| Northwest Territories | *Picea glauca* | 0.09 ± 0.11 | 0.79 ± 0.90 | 0.36 ± 0.62 | 0.50 ± 0.83 | 1.74 ± 1.63 |
| | *Picea mariana* | 0.09 ± 0.12 | 0.95 ± 1.09 | 0.39 ± 0.68 | 0.55 ± 0.91 | 1.99 ± 1.86 |
| Saskatchewan | *Larix laricina* | 0.08 ± 0.10 | 0.60 ± 0.68 | 0.33 ± 0.57 | 0.68 ± 1.13 | 1.68 ± 1.67 |
| | *Picea glauca* | 0.09 ± 0.12 | 0.80 ± 0.91 | 0.35 ± 0.60 | 0.43 ± 0.71 | 1.66 ± 1.55 |
| | *Picea mariana* | 0.09 ± 0.11 | 0.80 ± 0.92 | 0.37 ± 0.63 | 0.51 ± 0.84 | 1.76 ± 1.65 |


## Appendix B

**Table B1. Values of the exponent $c$ used in the weighted nonlinear regressions of component biomass against diameter at breast height (DBH) in which the weightings were inversely proportional to $DBH^{2c}$.** The number of observations ($n$) is provided for each regression.

| Biomass component | Magadan | | Ust-Yansky | |
|---|---|---|---|---|
| | $n$ trees | $c$ | $n$ trees | $c$ |
| Stem | 43 | 2 | 20 | 2.5 |
| Stem wood | 32 | 2 | 20 | 2.5 |
| Stem bark | 32 | 2.5 | 20 | 2 |
| Branches | 43 | 2 | 20 | 1.5 |
| Foliage | 43 | 1.5 | 20 | 1.5 |
| Aboveground | 43 | 2 | 20 | 2 |

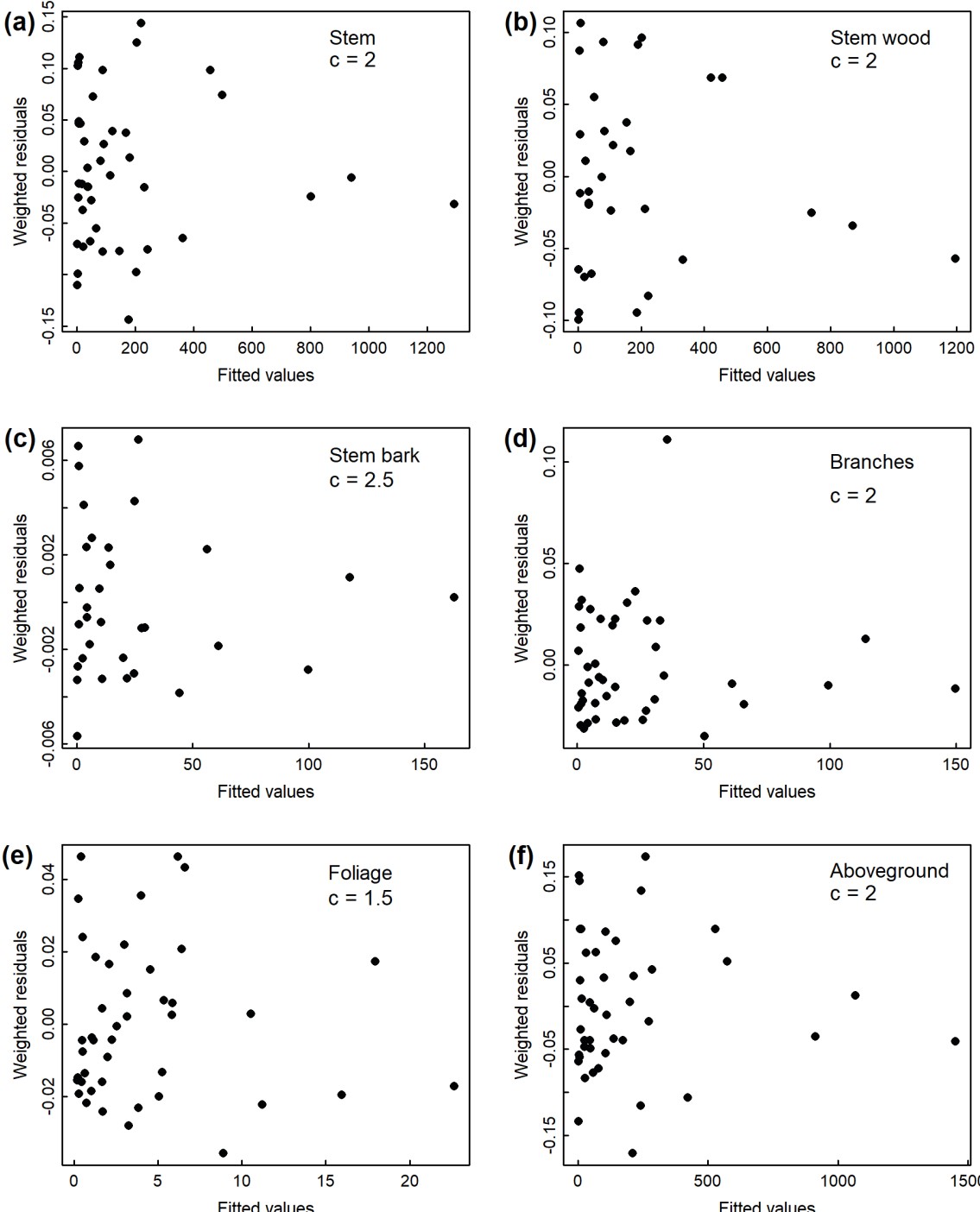

**Figure B1. Weighted residuals against fitted values from the weighted nonlinear regressions relating diameter at breast height (DBH) to (a) stem, (b) stem wood, (c) stem bark, (d) branches, (e) foliage, and (f) total aboveground tree biomass in the Magadan site.** Weightings were inversely proportional to $DBH^{2c}$.


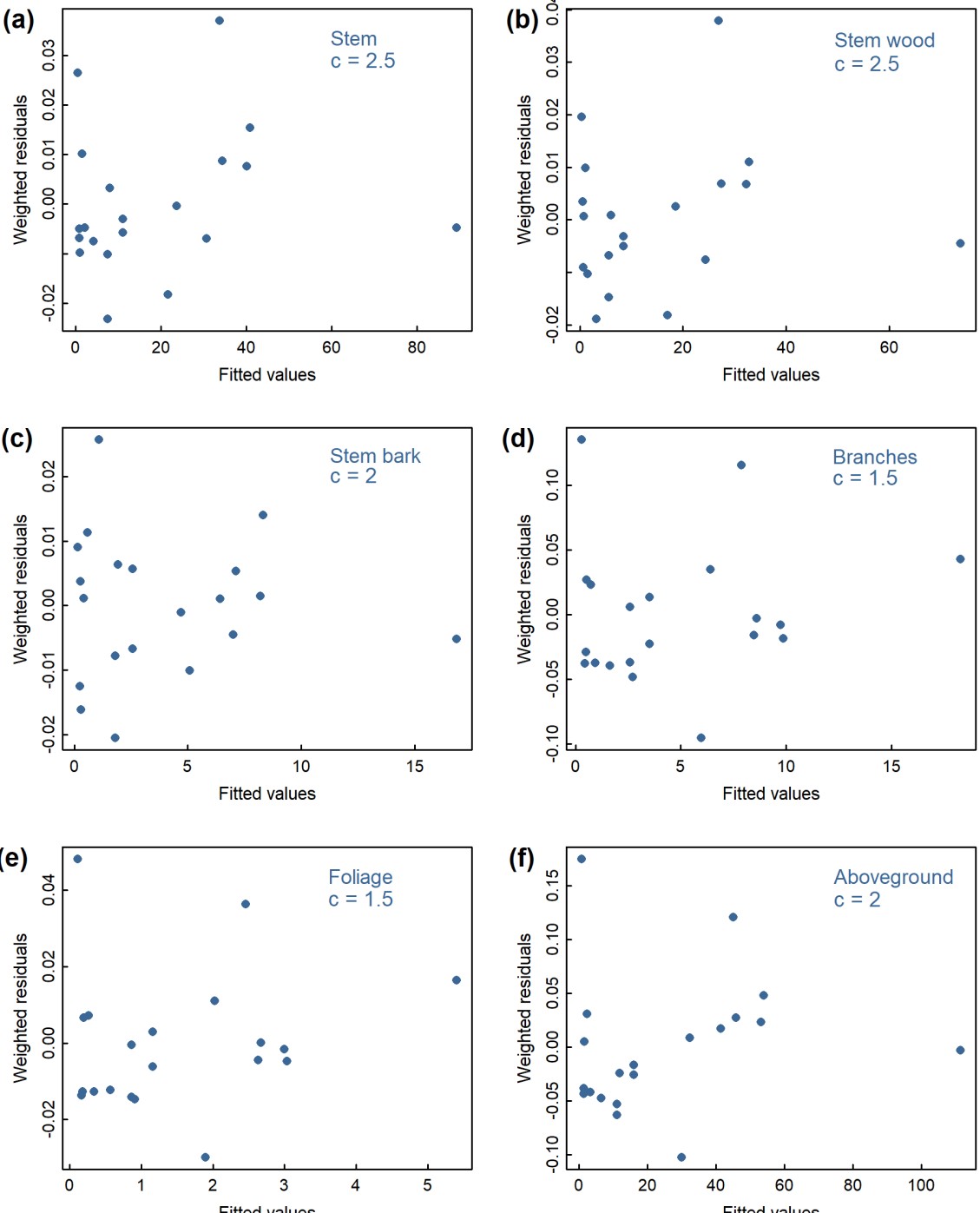

**Figure B2. Weighted residuals against fitted values from the weighted nonlinear regressions relating diameter at breast height (DBH) to (a) stem, (b) stem wood, (c) stem bark, (d) branches, (e) foliage, and (f) total aboveground tree biomass in the Ust-Yansky site.** Weightings were inversely proportional to $DBH^{2c}$.

## Appendix C

**Table C1. Coefficients of site-specific equations relating diameter at breast height (DBH) to stem, stem wood, stem bark, branches, foliage, and total aboveground biomass derived using linear regressions.** Equations are of the form $Y = a \times \text{DBH}^b$ where units of $Y$ are in kilograms of dry weight and DBH is in centimeters. A logarithmic transformation of the form $\ln Y = \ln a + b \times \ln \text{DBH}$ was applied before fitting the linear regression using the ordinary least squares method. Each $a$ value is corrected by the multiplication factor ($CF$). The coefficient of determination ($R^2$) is provided for each regression, as well as the root mean square error (RMSE). Differences in $a$ and $b$ coefficients among site-specific regressions are shown by $t$-values with significance levels (ANOVA with interaction term). All regressions are significant at $p < 0.001$. $ns$: not significant at $p < 0.05$, [*] $p < 0.05$, [**] $p < 0.01$, [***] $p < 0.001$.

| Component $Y$ | Coefficients | Site-specific allometry | | |
|---|---|---|---|---|
| | | Magadan | Ust-Yansky | $t$-value |
| Stem (kg) | $a$ | 0.045 | 0.123 | 3.288[**] |
| | $b$ | 2.637 | 2.251 | -2.657[*] |
| | $CF$ | 1.093 | 1.018 | |
| | $R^2$ | 0.950 | 0.987 | |
| | RMSE | 55.286 | 5.600 | |
| Stem wood (kg) | $a$ | 0.027 | 0.081 | 3.303[**] |
| | $b$ | 2.781 | 2.326 | -2.968[**] |
| | $CF$ | 1.099 | 1.026 | |
| | $R^2$ | 0.955 | 0.982 | |
| | RMSE | 101.133 | 5.392 | |
| Stem bark (kg) | $a$ | 0.005 | 0.045 | 5.840[***] |
| | $b$ | 2.664 | 2.020 | -3.807[***] |
| | $CF$ | 1.123 | 1.032 | |
| | $R^2$ | 0.941 | 0.972 | |
| | RMSE | 10.485 | 0.815 | |
| Branches (kg) | $a$ | 0.029 | 0.112 | 3.044[**] |
| | $b$ | 2.171 | 1.724 | -2.053[*] |
| | $CF$ | 1.215 | 1.054 | |
| | $R^2$ | 0.854 | 0.936 | |
| | RMSE | 17.110 | 1.749 | |
| Foliage (kg) | $a$ | 0.011 | 0.044 | 3.249[**] |
| | $b$ | 1.981 | 1.634 | -1.633 $ns$ |
| | $CF$ | 1.203 | 1.052 | |
| | $R^2$ | 0.837 | 0.933 | |
| | RMSE | 2.655 | 0.558 | |
| Aboveground (kg) | $a$ | 0.080 | 0.260 | 4.459[***] |
| | $b$ | 2.497 | 2.064 | -3.616[***] |
| | $CF$ | 1.058 | 1.023 | |
| | $R^2$ | 0.964 | 0.980 | |
| | RMSE | 50.395 | 5.561 | |

**Table C2. Coefficient of determination ($R^2$) from linear regressions between biomass components ($Y$) and size parameters ($D$) after log-log transformation.** All regressions are significant at $p < 0.001$. DBH: diameter at breast height (cm); $H$: tree height (m).

| Component $Y$ | Size parameter $D$ | Coefficient of determination ($R^2$) | |
|---|---|---|---|
| | | Magadan | Ust-Yansky |
| Stem (kg) | DBH (cm) | 0.950 | 0.987 |
| | DBH$^2{\times}H$ (cm$^2$ m) | 0.979 | 0.992 |
| Stem wood (kg) | DBH (cm) | 0.955 | 0.982 |
| | DBH$^2{\times}H$ (cm$^2$ m) | 0.983 | 0.989 |
| Stem bark (kg) | DBH (cm) | 0.941 | 0.972 |
| | DBH$^2{\times}H$ (cm$^2$ m) | 0.972 | 0.969 |
| Branches (kg) | DBH (cm) | 0.854 | 0.936 |
| | DBH$^2{\times}H$ (cm$^2$ m) | 0.870 | 0.938 |
| Foliage (kg) | DBH (cm) | 0.837 | 0.933 |
| | DBH$^2{\times}H$ (cm$^2$ m) | 0.861 | 0.935 |
| Aboveground (kg) | DBH (cm) | 0.964 | 0.980 |
| | DBH$^2{\times}H$ (cm$^2$ m) | 0.990 | 0.986 |

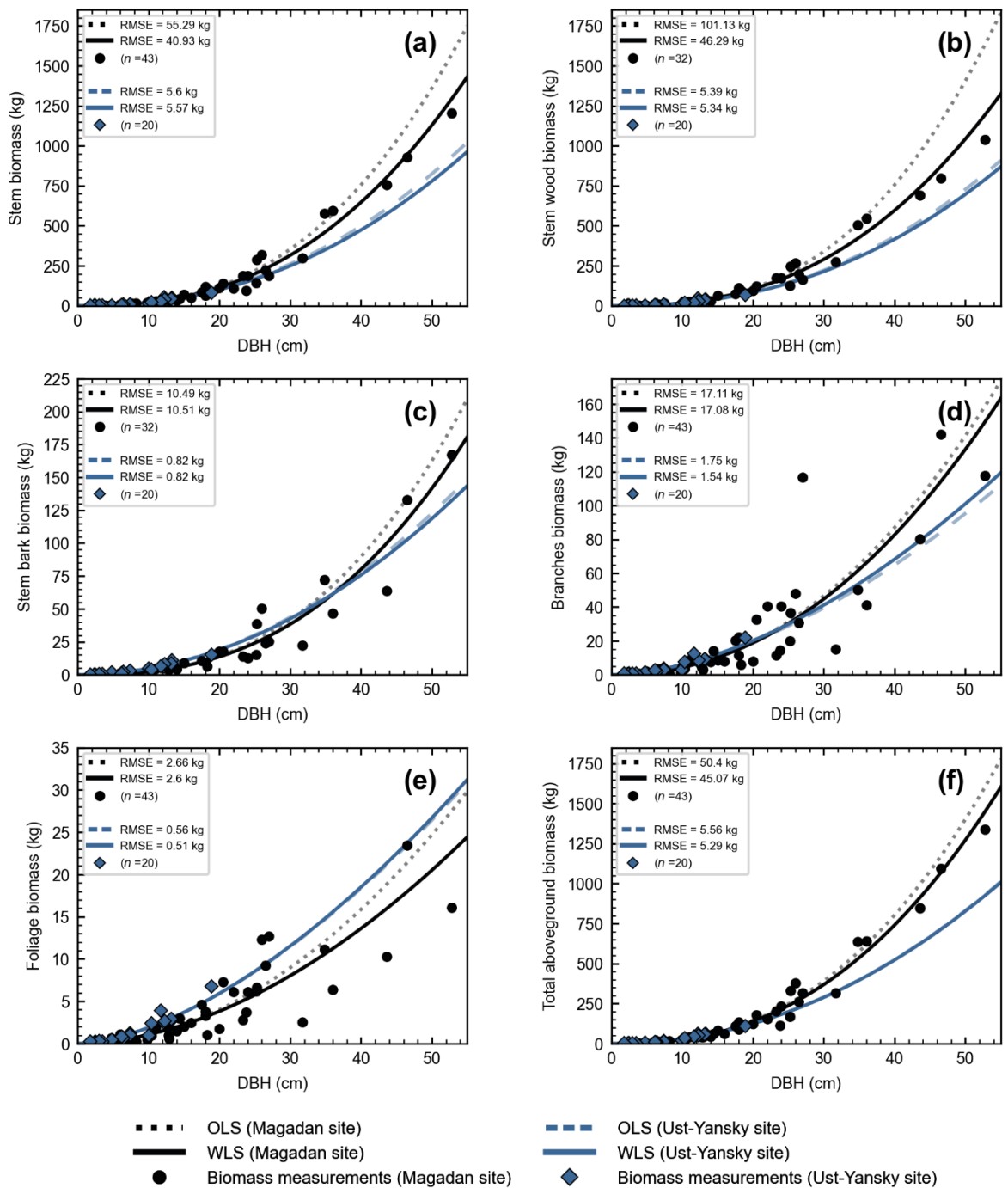

**Figure C1. Site-specific allometry models for *Larix cajanderi* developed using linear regressions (OLS) and weighted nonlinear regressions (WLS).** The markers (circles and diamonds) represent biomass measurements collected in seven forest stands in the Magadan Oblast and two forest stands in the Ust-Yansky district, with *n* the number of harvested trees reported in brackets. DBH: diameter at breast height; OLS: ordinary least squares; WLS: weighted least squares; RMSE: root mean square error.

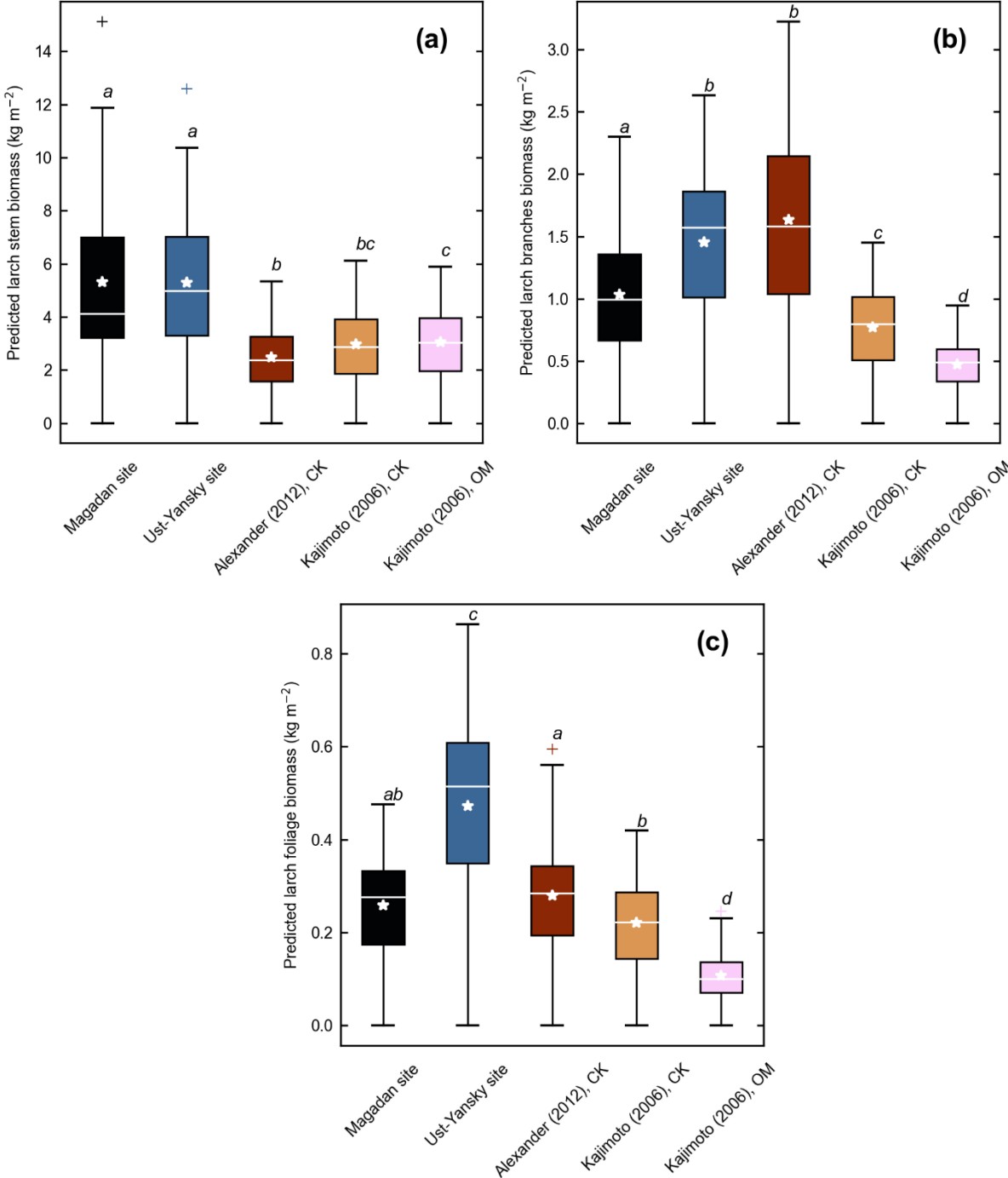

**Figure C2. Component biomass predicted for 53 forest stands near Yakutsk using allometry models developed in the Magadan**
**Oblast, the Ust-Yansky district, Chersky (CK) area (Kajimoto et al., 2006; Alexander et al., 2012) and Oymyakon (OM) area**
**(Kajimoto et al., 2006). (a) Stem, (b) branches, and (c) foliage biomass.** For each allometry model, mean predicted biomass is indicated
as a white star. Letters represent significant differences ($p < 0.05$) between allometry models determined using the Wilcoxon rank sum test
for pairwise comparisons.

**Appendix D**

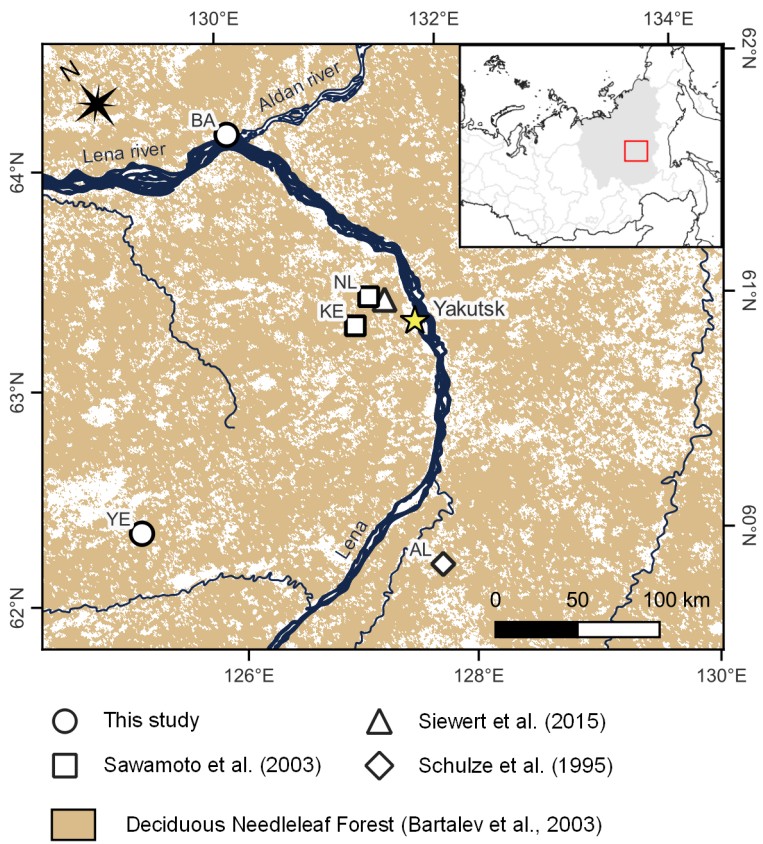


**Figure D1. Location of allometry-derived aboveground biomass estimates in larch-dominated forests around Yakutsk in the Republic of Sakha, Russia.** AL: Aldan Plateau; BA: Batamay; KE: Kenkeme; NL: Neleger; YE: Yert.

**Author contributions**

CJFD designed the research with inputs from SV. CJFD organized the field campaign with inputs from SV and the people in the Acknowledgements. CJFD performed the analysis, created figures, and wrote the manuscript with inputs from SV. SV acquired funding for this research and supervised the work.

**Competing interests**

The authors declare that they have no conflict of interest.

**Acknowledgements**


This work was supported by funding from the Dutch Research Council (NWO) through a Vidi grant (Fires Pushing Trees North, 016.Vidi.189.070) awarded to S.V. We would like to thank T. Maximov and R. Petrov for logistical and field support. We wish to thank B. Izbicki, B. M. Rogers, R. C. Scholten, T. A. Shestakova, and D. van Wees for contributing to data collection. We are grateful to Hans Cornelissen and Richard van Logtestijn for discussions on woody debris and tree cores
sampling. We are thankful to Ute Sass-Klaassen and Linar Akhmetzyanov for their inputs on dendrochronology. We thank D. Schepaschenko for compiling a comprehensive dataset of biomass measurements for Eurasia and making this data available.

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
