# Peer review of "Allometric equations and wood density parameters for estimating aboveground and woody debris biomass in Cajander larch (*Larix cajanderi*) forests of Northeast Siberia"

_Biogeosciences, 2022_

## Author Comment (AC1)

**AC1 – Reply on RC1 (Anonymous Referee #1)**

We respond below with original reviewer text in **black**, author comments in **blue**, and manuscript amendments given in **green**.

Dear. Editor

The current paper aims to parameterize i) allometric questions for aboveground biomass using an existing database and ii) an equation to estimate woody debris on the forest floor using data based on field survey, for Siberian *Larix cajanderi*. Authors demonstrate significant spatial variations of biomass of standing trees and woody debris estimates on the forest floor, depending on equations and parameters, comparing their estimated parameters with published ones. They come to the conclusion that the developed functions can be applicable to the species in Siberian forests. I consider that the paper would fail to fit within the journal's scope as well as may fail to attract a broad readership, because i) the study develops a tool, with no application of the tool, thus to fail to draw geo- or biological conclusion, ii) even the developed tools are only appliable at a relatively small scale, as no testing for the feasibility of a large scale application was made, while such tools for a large scale application already exist from national to continental scales, and iii) the estimated parameters were unjustly compared with published parameters, as to overfitting and comparing between different population distributions; for example, published equations were based on smaller and older trees, compared to the data set, based on which allometric questions were developed.

1A. We respectfully disagree with most elements of this assessment.

First, we do agree with the reviewer that pooling data from sites with different forest structure and stand age may not be desirable. We will therefore remove the site-common allometric equation from our revised paper.

We refer to our general response above as to why we believe that our paper fits well into the scope of Biogeosciences and why we are convinced that this is an important and timely contribution.

It is unclear what the reviewer refers to with overfitting as we have used state-of-the art statistical techniques that are commonly used to develop allometric equations. If the reviewer can explain in more detail what is meant with this comment, we would be happy to elaborate on this.

Here come specific comments.

L 15. "… at breast height (DBH)"

Depending on regions, the breast height differs. Specify the height (m).

1B. This is nominally at 1.3 m above the ground. We will add this is in the revision.

L51. "The line-intersect…"

Authors may begin a new paragraph before "The line- …"

1C. Thank you. We will do this in the revision.

2.1 Fine woody debris sampling

This section may be expanded and articulated. For example, papers that have been cited here (Sackett 1980; Van Wagner 1982; Nadel et al. 1997; 1999) articulate the formulation. Because the formulation is of great importance in the paper, it has to be well explained, and readers would not want to check back those papers to understand the formulation and meanings of parameters.

1D. Thank you for your comment. We will expand and articulate this session in the revision as follows:

L100: The line-intersect method is a widely used approach to quantify fine woody debris lying on the ground in a forest stand (Warren and Olsen, 1964; Van Wagner, 1968; Brown, 1971). It requires measuring the diameter of each piece of wood at its intersection with a sample line which can be considered as a strip of infinitesimal width containing a series of cross-sectional areas (Van Wagner, 1982). The sum of cross-sectional areas divided by the length of the sample line can then be converted to volume by multiplying both numerator and denominator by width. Fuel load is then obtained from Equation (1) by multiplying the volume by the specific gravity of wood as follows (Van Wagner, 1982):

$$W = \frac{\pi}{2} \times \sum d^2 \times \frac{\pi}{4} \times \frac{G}{L}, \tag{1}$$

where $W$ is fuel load or weight per unit ground area, $\pi/2$ is a probability factor that allows to sum the cross-sectional areas as circles, $d$ is piece diameter, $\pi/4$ is the factor required to convert $d^2$ into a circular area, $L$ is length of sample line, and $G$ is specific gravity in units of weight per unit volume. Equation (1) assumes that woody pieces are horizontal and does not account for ground slope. To minimize the bias related to tilted pieces that are less likely to be intercepted by the sample line, $W$ can be multiplied by a correction factor equal to the secant of the piece tilt angle relative to horizontal (Brown and Roussopoulos, 1974). Similarly, a correction factor can be calculated from the ground slope angle as follows (Brown, 1974):

$$s = \sqrt{1 + (\tan slope)^2}, \tag{2}$$

where $s$ is slope correction factor, slope is ground slope (degrees). Consequently,

$$W = \frac{\pi^2 \times G \times \sec h \times \sum d^2 \times s}{8 \times L}, \tag{3}$$

where $h$ is piece tilt angle (degrees). Measuring diameter on each intersected piece along a sample line can be tedious and time-consuming, especially if small pieces are abundant. In practice, FWD are tallied by diameter size class using a go/no-go sizing gauge, and the number of intercepts over the sample line is reported for each class (Brown, 1974; McRae et al., 1979). Therefore, the term $\sum d^2$ in

Equation (3) is replaced by $\sum_i n_i \times D_i^2$, where $n_i$ is the number of intercepts over the sample line in the diameter size class i, and $D_i$ is the representative class diameter (Van Wagner, 1982). The quadratic mean diameter (QMD) is generally used as the appropriate class diameter so that fuel load for any species and diameter size class i can be calculated as follows (Van Wagner, 1982; Nalder et al., 1999):

$$W_i = \frac{\pi^2 \times G_i \times \sec h_i \times n_i \times QMD_i^2 \times s}{8 \times L},$$ (4)

where $W_i$ is the fuel load (t ha$^{-1}$) for the diameter size class i, $G_i$ is the specific gravity (Mg m$^{-3}$) of the size class i, $h_i$ is the piece tilt angle (degrees) of the size class i, $n_i$ is the number of intercepts over the sample line in the size class i, s is the slope correction factor, L is the length of the sample line (m), and $QMD_i$ is the quadratic mean diameter (cm) of the size class i given by

$$QMD_i = \sqrt{\frac{d_i^2}{n_i}},$$ (5)

Brown, J. K: A Planar Intersect Method for Sampling Fuel Volume and Surface Area, For. Sci., 17, 96–102, doi:10.1093/forestscience/17.1.96, 1971.

Brown, J. K.: Handbook for inventorying downed woody material, U.S. Dept. of Agriculture, Forest Service, Intermountain Forest and Range Experiment Station, Ogden, Utah, Gen. Tech. Rep. INT-16, 24 pp., 1974.

Brown, J. K. and Roussopoulos, P. J.: Eliminating biases in the planar intersect method for estimating volumes of small fuels, For. Sci., 20, 350–356, 1974.

McRae, D. J., Alexander, M. E., and Stocks, B. J.: Measurement and description of fuels and fire behavior on prescribed burns: A handbook, Canadian Forestry Service, Great Lakes Forest Research Centre, Sault Ste. Marie, Ontario, Information Report O-X-287, 44 pp., 1979.

Nalder, I. A., Wein, R. W., Alexander, M. E., and de Groot, W. J.: Physical properties of dead and downed round-wood fuels in the Boreal forests of western and Northern Canada, Int. J. Wildl. Fire, 9, 85–99, doi:10.1071/WF00008, 1999.

Van Wagner, C. E.: The Line Intersect Method in Forest Fuel Sampling, For. Sci., 14, 20–26, 605 doi:10.1093/forestscience/14.1.20, 1968.

Van Wagner, C. E.: Practical aspects of the line intersect method, Canadian Forestry Service, Maritimes Forest Research Centre, Fredericton, New Brunswick, Information Report PI-X-12E, 11 pp., 1982.

Warren, W. G. and Olsen, P. F.: A Line Intersect Technique for Assessing Logging Waste, For. Sci., 10, 267–276, doi:10.1093/forestscience/10.3.267, 1964.

L130. Volume of a sample was first dried and then estimated. Would it underestimate volume of the sample due to shrinkage during drying?

1E. Thank you for your question. We followed the standard procedure (ASTM International, 2014). Yes, the sample may slightly shrink because of drying. We are interested in the volume of the dry matter (thus without moisture influences) and this procedure is consistently applied and reported within and across studies (e.g., Nalder et al., 1999).

ASTM International: ASTM D2395-14: Standard test methods for density and specific gravity (relative density) of wood and wood-based materials, ASTM International, West Conshohocken, Pennsylvania, United States, doi:10.1520/D2395-14, 2014.
Nalder, I. A., Wein, R. W., Alexander, M. E., and de Groot, W. J.: Physical properties of dead and downed round-wood fuels in the Boreal forests of western and Northern Canada, Int. J. Wildl. Fire, 9, 85–99, doi:10.1071/WF00008, 1999.

L 305. Figure 3.
I would recommend to add data points in the figures. Ranges of the independent variable
1F. Thank you. We agree and we will do this in revision.

---

## Author Comment (AC2)

**AC2 – Reply on RC2 (Anonymous Referee #2)**

We thank the reviewer for the constructive and valuable assessment of our paper. We respond below with original reviewer text in **black**, author comments in **blue**, and manuscript amendments given in **green**.

The authors of Allometric equations and wood density parameters for estimating aboveground and woody debris biomass in Cajander larch (Larix cajanderi) forests of Northeast Siberia observed mean squared diameter and specific gravity. They developed allometric equations at 25 sites in the Republic of Sakha in Russia. They then make comparisons to allometric equations developed in other studies. Overall the methods appear rigorous. This study provides valuable information from an important yet remote region of the world from which little in-situ data is available and makes the case that further data collection efforts are needed. My major concern is that the claim that the allometric equations presented in the text are more generally applicable needs to be better supported or more nuanced. The equations and the comparison of them to other previously developed equations are not validated against in situ data or across a larger region of space. It seems that the equations from earlier work would be more applicable than those developed by the authors in some more northeastern regions or specific stand types, especially near where they were developed. This is very problematic given the manuscripts focus on providing generalized equations to improve our ability to estimate above-ground biomass in this region.

2A. We agree with the reviewer that pooling data from sites with different forest structure and stand age may not be desirable. We will therefore remove the site-common allometric equation from our revised paper. By doing so, a main contribution of our paper is significantly increasing (from 3 to 5) the number of allometric equations available for Cajander larch forests. We agree with the reviewer that near the sampling sites of these five locations (two previous studies, two newly developed equations), the recommendation is to use the corresponding site-specific equation. A practical challenge arises when scientists would want to measure tree biomass in areas that are further away from those four sampling locations. In this case, we recommend that researchers could use site-specific equations when site parameters (e.g., stand age, DBH distribution, etc.) closely match with one of the sampling sites. When this is not the case, we recommend that researchers calculate tree biomass from the five different equations which will result in uncertainty range (e.g., mean biomass and standard deviation). These planned changes in the revision alter some of the main conclusions of the paper, for example those focused on generalized applicability, and we will rewrite these sections in order to reflect these changes.

L95 introduction: I suggest highlighting some of the other roles that this data could play (previously mentioned around L35) to strengthen the introduction.

2B. Thank you for your comment. We will do this in the revision.

L140: Is there a citation for the two equations above. Also, some brief explanation of why the samples were dipped in paraffin could be useful.

2C. Thank you for the comment. We will add the appropriate reference and specified why the samples were dipped in paraffin in the revision as follows:

L131: "After drying, the water displacement method requires to seal the surface of woody pieces to prevent bias in volume determination resulting from water absorption during immersion. Each oven-dried sample was then covered with a thin impermeable layer by immersion in hot liquid paraffin (solidification point 57–60 °C, 0.90 g cm$^{-3}$ at 20 °C), and the mass of the coated piece was measured again before volume determination."

L136: "Finally, specific gravity of each sample was determined as follows (ASTM International, 2014):

$$G = \frac{K \times m_0}{V_0},\qquad (3)$$

where $G$ is the specific gravity (g cm$^{-3}$), $K$ is a constant equal to 1 when the mass is in grams and the volume is in cubic centimeters, $m_0$ is the oven-dry mass (g) and $V_0$ is the oven-dry volume (cm$^3$), given by:

$$V_0 = m_{w,disp} - \frac{m_{coated} - m_0}{\rho_{paraffin}},\qquad (4)$$

where $m_{w,disp}$ is the mass of water displaced (g), $m_{coated}$ is the mass of the oven-dried sample after immersion in the paraffin (g), and $\rho_{paraffin}$ is the density of the paraffin wax (g cm$^{-3}$)."

L205-210: I suggest including a bit more information about this fitting method and selecting the exponent c, including references to other work that uses this method. It could also be useful to include these residual plots in the appendix.

2D. Thank you for your comment. We will include new references to other studies that use weighted nonlinear regression as a fitting method (e.g., Whraton and Cunia, 1987; Brown et al., 1989; Parresol, 1999; Moore, 2010) in the revision. We will explain in more detail how we selected the exponent *c* by (1) trial and error after visualization of the plots of the weighted residuals against the fitted values,

and (2) approximation of the conditional variance of biomass following the steps described below (Picard et al., 2012):

1. We divided DBH range into $K$ classes centered on $DBH_k$ ($k = 1,…, K$). We took $K = 5$ in this study and visually checked that the power model was appropriate for modeling the residual variance.

2. We calculated the empirical variance of biomass, $\sigma_k^2$, for the observations in class $k$ ($k = 1,…, K$)

3. We fitted the linear regression between $\ln(\sigma_k)$ and $\ln(DBH_k)$. The slope of this regression is an approximation of the exponent $c$.

We will specify that we used both approaches to estimate the exponent $c$ for each site (Yakutia and Magadan) and biomass component. We will include residual plots as supplementary materials in the revision of our paper.

Brown, S., Gillespie, A. J. R., and Lugo, A. E.: Biomass estimation methods for tropical forests with applications to forest inventory data, Forest Sci., 35, 881–902, doi:10.1093/forestscience/35.4.881, 1989.

Moore, J. R.: Allometric equations to predict the total above-ground biomass of radiata pine trees, Ann. For. Sci., 67, 806, doi:10.1051/forest/2010042, 2010.

Parresol, B. R.: Assessing tree and stand biomass: a review with examples and critical comparisons, Forest Sci., 45, 573–593, doi:10.1093/forestscience/45.4.573, 1999.

Picard, N., Saint-André, L., and Henry, M.: Manual for building tree volume and biomass allometric equations: from field measurements to prediction, Food and Agricultural Organization of the United Nations, Rome, Italy, and Centre de Coopération Internationale en Recherche Agronomique pour le Développement, Montpellier, France, 215 pp., 2012.

Whraton, E. H. and Cunia, T.: Estimating tree biomass regressions and their error, in: Proceedings of the workshop on tree biomass regression functions and their contribution to the error of forest inventory estimates, Syracuse, NY, USA, 26–30 May 1986, U.S. Dept. of Agriculture, Forest Service, Northeastern Forest Experiment Station, Broomall, Pennsylvania, USA, Gen. Tech. Rep. NE-117, 1987.

Figure 3: It would be good to include the data points on this plot where possible and the standard error envelopes for the fits. These lines are also somewhat difficult to distinguish when printed in black and white.

2E. Thank you for this comment. We will modify Figure 3 as suggested and will change the colors to improve its readability in gray scale in our revision.

L206: I think table A1 is important and merits inclusion in the text. It could be interesting to see this comparison done differently. For example, calculating fuel loads at one of the study sites using these different parameters and then plotting the values could better illustrate their importance in percentage terms.

2F. Thank you for this suggestion. We calculated fuel loads at 47 of 53 studied sites where *Larix cajanderi* fine woody debris were observed using *M* factors from this study. For each site, we derived the percentage difference between these estimates and fuel loads computed using *M* factors from other species and boreal regions provided in Nalder et al. (1999), and plotted this in a new figure as follows:

[Figure]

Figure. Percentage difference in fine woody debris (FWD) biomass estimates in 47 larch forest stands (*Larix cajanderi*) near Yakutsk using *M* factors derived for other species and regions. Differences were calculated from the estimates based on the *M* values developed in this study, such that a positive percentage difference reflects a lower biomass estimate. Each box ranges from the first quartile (Q1) to the third quartile (Q3), with the median and mean indicated by a white horizontal line and a star respectively. The whiskers extend from Q1 and Q3 to the minimum and maximum defined as Q1−1.5×IQR and Q3+1.5×IQR respectively, where IQR is the interquartile range (Q3−Q1). Outliers above the maximum or below the minimum are indicated by crosses.

L310: The Magadan site has many more samples than the Yakutian site, although the size range of the available samples varies. Given the conclusion that allometry is somewhat region-specific, it could be interesting to see the result of a fit where this imbalance of samples is corrected using weighting.

2G. That is a good suggestion. Based on the comments of reviewer 1 and you, we have decided to remove the site-common allometry from the revision, and now focus on providing two newly developed allometric equations and their potential use.

Figure 4: For this box plot of the site observations, it would be good to explain the quantiles, etc. shown in the figure caption

2H. Thank you for this suggestion. We will explain the statistics displayed in the boxplots in the figure caption in our revision as in the following: 'Each box ranges from the first quartile (Q1) to the third quartile (Q3), with the median and mean indicated by a white horizontal line and a star respectively. The whiskers extend from Q1 and Q3 to the minimum and maximum defined as Q1−1.5×IQR and Q3+1.5×IQR respectively, where IQR is the interquartile range (Q3−Q1). Outliers above the maximum or below the minimum are indicated by crosses.'

L315: Some additional summary information for these 53 sites could be useful (i.e., the mean, sd, and range of dbh)

2I. We agree that this information could be useful. It will be included in a new table that will summarize main characteristics of the 53 forest stands where we estimated FWD and tree biomass using the parameters and equations developed in this study.

**Table 5. Stand characteristics of 53 study sites located near Yakutsk (Republic of Sakha, Russia).** G: wood specific gravity; MSD: mean squared diameter; DBH: diameter at breast height.

| Forest stands characteristics | Site location | |
| --- | --- | --- |
| | **Batamay** | **Yert** |
| Latitude | 63°31'N | 62°01'N |
| Longitude | 129°23'E | 125°47'E |
| Number of stands | 30 | 23 |
| *Number of sites with measurements/estimates for* | | |
| Fine woody debris, *G* and MSD | 4 | 21 |
| Fine woody debris biomass | 25 | 22 |
| DBH, tree biomass | 30 | 23 |
| *Larch trees* | | |
| Mean density ± std (range) (thousand trees ha$^{-1}$) | 10.55 ± 9.93 (1.00–45.33) | 13.50 ± 13.69 (0.17–46.33) |
| Mean tree age ± std (range) (years) | 83.2 ± 40.9 (9–214) | 97.1 ± 39.1 (40–162) |
| Mean DBH ± std (range) (cm) | 5.48 ± 3.56 (1.20–15.08) | 5.09 ± 5.14 (0.22–18.56) |

L315-320, 350-365: I suggest heavily revising these sections of the paper. The claim that the other allometric equations underestimate aboveground biomass or are more generally applicable seems too strong. The actual aboveground biomass of the 53 comparison sites is not truly known. From the text or maps, it's unclear where exactly these test sites are located and how close they are to the sites from Siewert 2015. The comparison to Siewet 2015 is not emphasized in the text. This point would be much stronger if some additional non-allometry-derived data sources, more information from the literature, etc., were included as validation. It could also be interesting to see this comparison done across multiple sites. I imagine these other two equations will perform better in certain areas or stand types. Such a comparison would add more depth to the point about how generalizable each set of equations is.

2J. Thank you for this comment. We agree with you. As explained in our response 2A, we decided to remove the site-common allometry, and focus on our two newly developed allometric equations and their potential use. This changes some of the main conclusions of our work and we will rewrite these sections accordingly.

The aim of comparing the different available equations (3 existing, 2 newly developed) was to show the differences in biomass estimates that result from using the currently available allometric relationships for Eastern Siberian larch forests. We are aware that the estimates at our sampling plots do not represent ground truth. Yet, we believe that this comparison is of interest as it demonstrates that using the available site-specific allometric equations for Cajander larch forests may lead to significantly different biomass estimates. We already stated that further field efforts to advance our understanding of forest structure and biomass in Cajander larch is necessary and will further reinforce this point in the revision, yet we believe that our paper provides advances in this direction by providing new measurements (for fine woody debris) and increasing the number of available allometric equations.

Following your suggestion, we will include additional comparisons with studies that reported aboveground biomass estimates for larch-dominated forest stands in nearby areas. We will include a new map figure to show the locations of these study sites in reference to our 53 sampling locations.

L313-320: Figure 3d shows differences between our newly developed allometric models and existing aboveground biomass equations developed for *Larix cajanderi* trees in northeast Siberia (Kajimoto et al., 2006; Alexander et al., 2012). To illustrate these differences, we calculated aboveground biomass at 53 study sites near Yakutsk using each allometric model (Figure 1, new table from our response 2I). Total aboveground biomass estimated by applying site-specific allometry from the Ust-Yansky district (Yakutia) averaged at 7.21 ± 3.42 kg m$^{-2}$ (range: 0–15.1 kg m$^{-2}$), and at 6.58 ± 3.94 kg m$^{-2}$ (range: 0–

17.9 kg m$^{-2}$) when using the equation developed for the Magadan oblast (Fig. 4). Estimates from allometric relationships developed for the Chersky and Oymyakon areas resulted in significant lower biomass estimates (Wilcoxon, $p < 0.05$), ranging from 3.64 kg m$^{-2}$ to 4.40 kg m$^{-2}$ (Fig. 4). A similar trend was found when predicting stem biomass (Fig. 3; Fig. B2). Aboveground biomass is assumed to vary along a latitudinal gradient within larch forests of Northeast Siberia (Usoltsev et al., 2002), ranging from 0.1 kg m$^{-2}$ in northern regions to 18 kg m$^{-2}$ in more productive southern stands (Usoltsev, 2001; Kajimoto et al., 2010). Using the allometric equations developed in this study, mean aboveground biomass estimates across our larch forest stands were 33–48% higher than predictions from existing biomass equations. Our estimates were in good agreement with allometry-derived values reported for similar larch forest ecosystems located in the same area (new map figure). Indeed, Siewert et al. (2015) reported a mean aboveground tree biomass of 7.2 kg m$^{-2}$ in 12 *L. cajanderi* dominated forest stands in the Spasskaya Pad/Neleger study area (62°14'N, 129°37'E). Our study area was in the contact zone between *Larix cajanderi* and another closely related Siberian larch species, *Larix gmelinii* (Rupr.) Rupr. (Abaimov, 2010). Schulze et al. (1995) and Sawamoto et al. (2003) used allometric relationships to compute aboveground biomass in *L. gmelinii* forest stands. Their estimates ranged from 4.4 kg m$^{-2}$ (49 years old) to 12.0 kg m$^{-2}$ (125 years-old), and 2.0 kg m$^{-2}$ (25 years old) to 10.5 kg m$^{-2}$ (170 years-old). The biomass estimates at our study sites are of the same order as those of previous studies in nearby areas. Further efforts are needed to verify and validate the utility of our newly developed allometric equations. Part of this effort could focus on cross-comparing biomass estimates derived from allometric equations with remote sensing measurements from optical and light detection and ranging (LiDAR) sensors.

L339-347: Our study shows that using our newly developed allometric equations may result in higher biomass estimates compared to values derived from existing DBH-based equations. This may have important implications for understanding changes in boreal vegetation dynamics, carbon, and energy budgets in a warming climate. Our results suggest that site-specific allometric equations should be applied near the sampling sites from which they were developed. Given the vastness of the area covered by Cajander larch and the comparatively limited number of available allometric equations, a practical challenge arises when one would want to measure tree biomass in areas that are further away from the four sampling locations described in this study (i.e., the Ust-Yansky district, Chersky and Oymyakon areas in the Republic of Sakha, and the Magadan Oblast). In this case, we recommend using site-specific equations when site parameters (e.g., stand age, DBH, tree height, etc.) closely match with one of the sampling sites. When this is not the case, we recommend that one calculates

tree biomass from the five different equations (Fig. 4). This will result in a mean estimate and an uncertainty range.

L330-340: Interpreting the fitted allometric parameters (i.e., as in Niklas 1994) here and further discussing the differences in climate and other properties between the sites could strengthen the conclusions in this section.

2K. This is a good point, and we will include this interpretation and discussion in the revision. We will emphasize differences in climate, stand structure (e.g., tree density, stand age, DBH), permafrost characteristics, natural disturbances, that can explain differences between site-specific allometric relationships. We will include a summary of the tree samples in the Chersky and Oymyakon areas in Table 1 to facilitate the interpretation of differences in forest structure.

---

## Author Comment (AC3)

**AC3 – Reply on RC3 (Anonymous Referee #3)**

We thank reviewer 3 for the constructive and valuable assessment of our paper. We respond below with original reviewer text in **black**, author comments in **blue**, and manuscript amendments in **green**.

This manuscript describes the size-class-dependent characteristics of fine woody debris according to their detailed field experiment, and the new allometric relationship by using literature data, for the Cajander larch forest in Northeast Siberia. Since such data in this region (central Yakutia) is limited, this manuscript will contribute a lot to future modeling and remote sensing studies in this region. I've already seen the comments from the two reviewers and the author's replies, so I'd like to add some minor comments about what I'm still unsure about.

About fine woody debris: The authors show the MSD and specific gravity of Cajander larch for each diameter size class.

On the other hand, they compared their single factor $M$ with other species in different regions in Table A1, but it is shown in the percentage difference by size class, not the actual values of $M$. I think the single factor M and the fuel load W can be the important outcomes of this study, so I suggest the authors show these results.

3A. Thank you for this suggestion. We will include Table A1 in the main text and modify it to show the actual values of $M$ (see below). We will also include a new figure, as described in RC2, that shows the percentage difference in FWD fuel loads in 47 larch forest stands using $M$ factors derived for other species and regions. Differences were calculated from the estimates based on the $M$ values developed in this study (i.e., *Larix cajanderi* in Northeast Siberia). Finally, we will add a new table in appendix with fuel load means and ranges per diameter size class for each location and species.

**Table. Values of the multiplication factor $M$ from this study and those from other boreal tree species in the Canadian Northwest Territories and Saskatchewan by diameter size class.** $M$ values for *Larix laricina* (tamarack), *Picea glauca* (white spruce), and *Picea mariana* (black spruce) were derived from Equation (5) using specific gravity ($G$) and mean squared diameter (MSD) values from Nalder et al. (1999). To facilitate comparisons with $M$ values from our study, we used a tilt correction factor (sec h) of 1.13 as suggested by Brown (1974).

| Location | Species | Value of $M$ per diameter size class (II-V) | | | |
|---|---|---|---|---|---|
| | | II | III | IV | V |
| Northeast Siberia | *Larix cajanderi* | 0.628 | 2.89 | 12.1 | 32.3 |
| Northwest Territories | *Picea glauca* | 0.389 | 2.45 | 10.6 | 22.9 |
| | *Picea mariana* | 0.424 | 2.94 | 11.5 | 25.2 |
| Saskatchewan | *Larix laricina* | 0.338 | 1.85 | 9.63 | 31.1 |
| | *Picea glauca* | 0.397 | 2.47 | 10.1 | 19.5 |
| | *Picea mariana* | 0.380 | 2.48 | 10.7 | 23.1 |

[Figure]

Figure. Percentage difference in fine woody debris (FWD) biomass estimates in 47 larch forest stands (*Larix cajanderi*) near Yakutsk using $M$ factors derived for other species and regions. Differences were calculated from the estimates based on the $M$ values developed in this study, such that a positive percentage difference reflects a lower biomass estimate. Each box ranges from the first quartile (Q1) to the third quartile (Q3), with the median and mean indicated by a white horizontal line and a star respectively. The whiskers extend from Q1 and Q3 to the minimum and maximum defined as Q1−1.5×IQR and Q3+1.5×IQR respectively, where IQR is the interquartile range (Q3−Q1). Outliers above the maximum or below the minimum are indicated by crosses.

Equation (1):

- Even though the final answer is correct, I strongly suggest you adopt the consistent units in the equation. Specifically, the unit of QMD should be [m], not [cm], and the equation should be multiplied by $10^4$ to convert the unit from [Mg m$^{-2}$] to [Mg ha$^{-1}$]. This will avoid confusion by the readers and avoid careless mistakes in calculation.

3B. Thank you for your comment. We will express the specific gravity ($G$) in [g cm$^{-3}$] to be consistent throughout our manuscript. However, we think that QMD still need to be expressed in [cm] in the revision. Using both [m] and [cm] in Equation (1) might be confusing for the readers, yet this equation has originally been developed with the diameter of the woody piece ($d$) and the length of the transect line ($L$) expressed in [cm] and [m] (Van Wagner, 1968; 1982). Particularly, the quantity $\frac{\pi^2}{8}$ is a constant, usually referred to as $k$ (Van Wagner, 1982), that allows to retrieve fuel load [t ha$^{-1}$] from $d$ [cm] and $L$ [m]. This equation is consistently applied and reported within and across studies (e.g., Delisle and Woodard, 1988; Nalder et al., 1999; Alexander et al., 2004; Santín et al., 2015).

$$\frac{\left[\text{g cm}^{-3}\right] \times \left[\text{cm}\right]^2}{\left[\text{m}\right]} \Leftrightarrow \frac{\left[\text{g}\right] \times \left[\text{cm}\right]^{-1}}{100 \times \left[\text{cm}\right]} \Leftrightarrow \frac{\left[\text{g cm}^{-2}\right]}{100} \Leftrightarrow \frac{100 \times \left[\text{t ha}^{-1}\right]}{100} \Leftrightarrow \left[\text{t ha}^{-1}\right]$$

Alexander, M. E., Stefner, C. N., Mason, J. A., Stocks, B. J., Hartley, G. R., Maffey, M. E., Wotton, B. M.; Taylor, S. W., Lavoie, N., Dalrymple, G. N.: Characterizing the jack pine – black spruce fuel complex of the International Crown Fire Modelling Experiment (ICFME), Canadian Forestry Service, Northern Forestry Centre, Edmonton, Alberta, Inf. Rep. NOR-X-393, 2004.

Delisle G. P. and Woodard P. M.: Constants for calculating fuel loads in Alberta, Canadian Forestry Service, Northern Forestry Centre, Edmonton, Alberta, Forest Management Note No. 45, 1988.

Nalder, I. A., Wein, R. W., Alexander, M. E., and de Groot, W. J.: Physical properties of dead and downed round-wood fuels in the Boreal forests of western and Northern Canada, Int. J. Wildl. Fire, 9, 85–99, doi:10.1071/WF00008, 1999.

Santín, C., Doerr, S. H., Preston, C. M., and González-Rodríguez, G.: Pyrogenic organic matter production from wildfires: a missing sink in the global carbon cycle, Glob. Change Biol., 21, 1621-1633, doi:10.1111/gcb.12800, 2015.

Van Wagner, C. E.: The Line Intersect Method in Forest Fuel Sampling, For. Sci., 14, 20–26, 605 doi:10.1093/forestscience/14.1.20, 1968.

Van Wagner, C. E.: Practical aspects of the line intersect method, Canadian Forestry Service, Maritimes Forest Research Centre, Fredericton, New Brunswick, Information Report PI-X-12E, 11 pp., 1982.

- Secant (sec) should be in non-italic.

  3C. We will change this in the revision.

- Is $G_i$ the arithmetic mean of $G$ (specific gravity) within the diameter size class $i$?

  3D. $G_i$ is the specific gravity of the diameter size class $i$ for a given species and location. Our values are shown in Table 3. They were indeed derived by calculating the arithmetic mean of specific gravity within each size class. We will explicit this in the revision.

- Is $h_i$ the arithmetic mean of $h$ (piece tilt angle) within the diameter size class $i$? If yes, is it mathematically correct to calculate the secant using the arithmetic mean value of h for obtaining the fuel load?

  - For example, If $h$ takes 0 degrees and 180 degrees, the arithmetic mean of them can be 90 degrees.
  - Besides, according to Fig. 2, $h$ is always related to the diameter of each sampled piece, so I think the product of the diameter and sec $h$ should be used for the statistical calculation.

3E. Thank you for your comment. The basic equation developed by Van Wagner (1968) to retrieve fuel load using the line-intersect approach assumed that sampled pieces lie horizontally on the ground. If pieces are tilted, due to ground slope or because the piece is partially hanging, they are less likely to be intercepted by the transect line. h is the angle between the piece and the horizontal plane and is therefore not related to the diameter of the piece (h < 90°). To minimize the bias related to non-horizontal pieces, $W$ can be multiplied by a correction factor equal to the secant of the angle of tilt from horizontal (h) (Brown, 1974). Brown and Roussopoulos (1974) showed that the average correction factor for naturally fallen branches in American conifer forests ranged between 1.09 (h ≈ 23°) and 1.21 (h ≈ 34°). Tilted bias can be larger in fresh logging slash (correction factor as high as 1.38) where smaller pieces are attached to larger ones. $h_i$ is the specific tilt angle for the diameter size class $i$. It can be derived from the arithmetic mean of h within the size class $i$ (e.g., Nalder et al., 1999).

Brown, J. K.: Handbook for inventorying downed woody material, U.S. Dept. of Agriculture, Forest Service, Intermountain Forest and Range Experiment Station, Ogden, Utah, Gen. Tech. Rep. INT-16, 24 pp., 1974.

Brown, J. K. and Roussopoulos, P. J.: Eliminating biases in the planar intersect method for estimating volumes of small fuels, For. Sci., 20, 350–356, 1974.

Van Wagner, C. E.: The Line Intersect Method in Forest Fuel Sampling, For. Sci., 14, 20–26, 605 doi:10.1093/forestscience/14.1.20, 1968.

Nalder, I. A., Wein, R. W., Alexander, M. E., and de Groot, W. J.: Physical properties of dead and downed round-wood fuels in the Boreal forests of western and Northern Canada, Int. J. Wildl. Fire, 9, 85–99, doi:10.1071/WF00008, 1999.

- If $N$ represents the (total) number of intercepts over the length of the transect line, what does $N_i$ mean?

  3F. $N_i$ represents the number of intercepts per diameter size class $i$ over the length of the sample line. We will explicit this definition in the revision.

Equation (2):

- I suggest using a single character (e.g., α) instead of "*slope*" to represent the ground slope.

  3G. Thank you for this suggestion. We will change this in the revision.

- $(\tan \alpha)^2$ is generally written as $\tan^2\alpha$.

  3H. We will rewrite Equation (2) as suggested.

Equation (3) and L140:

- The authors use two characters to represent the specific gravity. One is $G$ in equation (1) and L104 (kg m$^{-3}$), and another is $S$ here (g cm$^{-3}$).

  3I. Thank you for this comment. We will use a single character to represent the specific gravity (i.e., $G$ in g cm$^{-3}$).

L158-159, equation (5):

- Does a single factor $M$ represent the fuel loads per intercept (sample) on the transect line? Please explain this concept concisely since the reference (Nalder et al., 1999) was not accessible from my environment.

3J. $M$ is a multiplication factor introduced by Nalder et al. (1999) to simplify fuel loads calculations. It combines the values of specific gravity, tilt angle, and MSD, as shown in Equation (5). For each diameter size class $i$, fuel loads are then obtained as follows:

$$W_i = \frac{N_i \times M_i}{L},$$

where $W_i$ is the fuel load (t ha$^{-1}$) for the diameter size class $i$, $N_i$ is the number of intercepts within the size class $i$, $M_i$ is the appropriate multiplication factor (g cm$^{-1}$) derived from $G_i$ (g cm$^{-3}$), $h_i$ (degrees), MSD$_i$ (cm$^2$), and $L$ is the length of the transect line (m). It represents the fuel load per intercept per meter of transect.

Nalder, I. A., Wein, R. W., Alexander, M. E., and de Groot, W. J.: Physical properties of dead and downed round-wood fuels in the Boreal forests of western and Northern Canada, Int. J. Wildl. Fire, 9, 85–99, doi:10.1071/WF00008, 1999.

- If you share the same units with equation (1), $G_i$ has the unit of [Mg m$^{-3}$], and MSD$_i$ might have the unit of [cm$^2$]. However, the author specified that $M$ has the unit of [g cm$^{-1}$]. In this case, the units of the left and right sides of equation (5) are inconsistent. I suppose the unit of $G_i$ in equation (5) would be [g cm$^{-3}$], or it should be $S_i$ according to equation (3).

  3K. Thank you for your comment. Mg m$^{-3}$ is equivalent to g cm$^{-3}$, but we will change the unit of $G_i$ as g cm$^{-3}$ in the revision so that both sides of Equation (5) have similar units.

- As pointed out in equation (1), I still wonder whether the use of "sec $h_i$" is mathematically correct if $h_i$ represents the arithmetic mean of $h$ in class $i$.

  3L. Please see our response 3E. The tilt correction factor (sec h) has consistently been applied and reported within and across studies (e.g., Brown and Roussopoulos, 1974; Nalder et al., 1999).

  Brown, J. K. and Roussopoulos, P. J.: Eliminating biases in the planar intersect method for estimating volumes of small fuels, For. Sci., 20, 350–356, 1974.

  Nalder, I. A., Wein, R. W., Alexander, M. E., and de Groot, W. J.: Physical properties of dead and downed round-wood fuels in the Boreal forests of western and Northern Canada, Int. J. Wildl. Fire, 9, 85–99, doi:10.1071/WF00008, 1999.

3M. Based on the comments of reviewer 1 and you, we will articulate the section *2.1 Fine woody debris sampling* as follows:

L100: The line-intersect method is a widely used approach to quantify fine woody debris lying on the ground in a forest stand (Warren and Olsen, 1964; Van Wagner, 1968; Brown, 1971). It requires

measuring the diameter of each piece of wood at its intersection with a sample line which can be considered as a strip of infinitesimal width containing a series of cross-sectional areas (Van Wagner, 1982). The sum of cross-sectional areas divided by the length of the sample line can then be converted to volume by multiplying both numerator and denominator by width. Fuel load is then obtained from Equation (1) by multiplying the volume by the specific gravity of wood as follows (Van Wagner, 1982):

$$W = \frac{\pi}{2} \times \sum d^2 \times \frac{\pi}{4} \times \frac{G}{L}, \tag{1}$$

where W is fuel load or weight per unit ground area, $\pi/2$ is a probability factor that allows to sum the cross-sectional areas as circles, d is piece diameter, $\pi/4$ is the factor required to convert $d^2$ into a circular area, L is length of sample line, and G is specific gravity in units of weight per unit volume. Equation (1) assumes that woody pieces are horizontal and does not account for ground slope. To minimize the bias related to tilted pieces that are less likely to be intercepted by the sample line, W can be multiplied by a correction factor equal to the secant of the angle between the piece and the horizontal plane (Brown and Roussopoulos, 1974). Similarly, a correction factor can be calculated from the ground slope angle as follows (Brown, 1974):

$$s = \sqrt{1 + \tan^2 \alpha}, \tag{2}$$

where s is slope correction factor, $\alpha$ is ground slope (degrees). Consequently,

$$W = \frac{\pi^2 \times G \times \sec h \times \sum d^2 \times s}{8 \times L}, \tag{3}$$

where h is piece tilt angle (degrees). Measuring diameter on each intersected piece along a sample line can be tedious and time-consuming, especially if small pieces are abundant. In practice, FWD are tallied by diameter size class using a go/no-go sizing gauge, and the number of intercepts over the sample line is reported for each class (Brown, 1974; McRae et al., 1979). Therefore, the term $\sum d^2$ in Equation (3) is replaced by $\sum_i n_i \times D_i^2$, where $n_i$ is the number of intercepts over the sample line in the diameter size class i, and $D_i$ is the representative class diameter (Van Wagner, 1982). The quadratic mean diameter (QMD) is generally used as the appropriate class diameter so that fuel load for any species and diameter size class i can be calculated as follows (Van Wagner, 1982; Nalder et al., 1999):

$$W_i = \frac{\pi^2 \times G_i \times \sec h_i \times n_i \times QMD_i^2 \times s}{8 \times L}, \tag{4}$$

where $W_i$ is the fuel load (t ha$^{-1}$) for the diameter size class i, $G_i$ is the specific gravity (g cm$^{-3}$) of the size class i, $h_i$ is the piece tilt angle (degrees) of the size class i, $n_i$ is the number of intercepts over the sample line within the size class i, s is the slope correction factor, L is the length of the sample line (m), and $QMD_i$ is the quadratic mean diameter (cm) of the size class i given by

$$QMD_i = \sqrt{\frac{\sum_i d_i^2}{n_i}}, \tag{5}$$

---

## Author Response (AR1)

We thank the three reviewers for their constructive and valuable assessments of our manuscript. We respond below with comments from reviewers in **black**, authors' responses in **blue**, and authors' changes in the manuscript in **green**.

**Responses to reviewer 1**

Dear. Editor

The current paper aims to parameterize i) allometric questions for aboveground biomass using an existing database and ii) an equation to estimate woody debris on the forest floor using data based on field survey, for Siberian *Larix cajanderi*. Authors demonstrate significant spatial variations of biomass of standing trees and woody debris estimates on the forest floor, depending on equations and parameters, comparing their estimated parameters with published ones. They come to the conclusion that the developed functions can be applicable to the species in Siberian forests. I consider that the paper would fail to fit within the journal's scope as well as may fail to attract a broad readership, because i) the study develops a tool, with no application of the tool, thus to fail to draw geo- or biological conclusion, ii) even the developed tools are only appliable at a relatively small scale, as no testing for the feasibility of a large scale application was made, while such tools for a large scale application already exist from national to continental scales, and iii) the estimated parameters were unjustly compared with published parameters, as to overfitting and comparing between different population distributions; for example, published equations were based on smaller and older trees, compared to the data set, based on which allometric questions were developed.

1A. We respectfully disagree with most elements of this assessment.
First, we do agree with the reviewer that pooling data from sites with different forest structure and stand age may not be desirable. We have therefore removed the site-common allometric equation from the revision.

Reviewer 1 asserts that our paper may not fall within the scope of Biogeosciences. We respectfully disagree with this assessment. On the webpage of Biogeosciences, the aims and scope of the journal states an interest in work on *'all aspects of the interactions between the biological, chemical, and physical processes in terrestrial or extraterrestrial life with the geosphere, hydrosphere, and atmosphere'* (https://www.biogeosciences.net/about/aims_and_scope.html).
Aboveground forest biomass and carbon storage are critical components at the interface of the biosphere and atmosphere, and further influence processes in the hydrosphere (e.g., through plantwater interactions), pedosphere (e.g., through nutrient cycling) and cryosphere (e.g., through influences on snow cover) among others.

The major contributions of our paper are (including the revisions):

- The first ever published wood density measurements that allow to calculate fine woody debris biomass in Cajander larch forests of Northeastern Siberia.

- A significant increase (from 3 to 5) in the number of available allometric equations to calculate tree biomass in Cajander larch forests of Northeastern Siberia. The previously published equations (Kajimoto et al., 2006; Alexander et al., 2012) do not include bark as a biomass pool. This is another important addition of our work, as quantifying bark biomass is important (e.g., bark and wood are characterized by different combustion processes; Lestander et al., 2012).

We want to stress that our contributions are focused on an extremely data-poor region, Northeastern Siberia larch forests, which encompass about 20 % of the boreal forest and are of global importance. Our paper provides a number of equations and parameters, in easily accessible manner, that will be of interest to anyone studying biomass and carbon stocks of Cajander larch forests in Siberia, and more broadly the boreal forest. We sincerely hope that more data on Cajander larch forests will become available in the future, so that this work can further be advanced. We have nuanced our statements with regards to the use of the data and equations in our paper and included this future perspective.

It is unclear what the reviewer refers to with overfitting as we have used state-of-the art statistical techniques that are commonly used to develop allometric equations. If the reviewer can explain in more detail what is meant with this comment, we would be happy to elaborate on this.

Alexander, H. D., Mack, M. C., Goetz, S., Loranty, M. M., Beck, P. S. A., Earl, K., Zimov, S., Davydov, S., and Thompson, C. C.: Carbon Accumulation Patterns During Post-Fire Succession in Cajander Larch (Larix cajanderi) Forests of Siberia, Ecosystems, 15, 1065–1082, doi:10.1007/s10021-012-9567-6, 2012.

Kajimoto, T., Matsuura, Y., Osawa, A., Abaimov, A. P., Zyryanova, O. A., Isaev, A. P., Yefremov, D. P., Mori, S., and Koike, T.: Size-mass allometry and biomass allocation of two larch species growing on the continuous permafrost region in Siberia, For. Ecol. Manage., 222, 314–325, doi:10.1016/j.foreco.2005.10.031, 2006.

Lestander, T. A., Lundström, A., and Finell, M.: Assessment of biomass functions for calculating bark proportions and ash contents of refined biomass fuels derived from major boreal tree species, Can. J. Forest Res., 42, 59–66, doi:10.1139/x11-144, 2012.

Here come specific comments.

L 15. "… at breast height (DBH)"
Depending on regions, the breast height differs. Specify the height (m).

1B. This is nominally at 1.3 m above the ground. We added this is in the revision.

L15–16 (p. 1): We developed allometric equations relating diameter at breast height (DBH at 1.3 m) …

L51. "The line-intersect…"
Authors may begin a new paragraph before "The line- …"

1C. We changed this in the revision.

2.1 Fine woody debris sampling
This section may be expanded and articulated. For example, papers that have been cited here (Sackett 1980; Van Wagner 1982; Nadel et al. 1997; 1999) articulate the formulation. Because the formulation is of great importance in the paper, it has to be well explained, and readers would not want to check back those papers to understand the formulation and meanings of parameters.

1D. We expanded this session and articulated the formulation of FWD loads as suggested.

L102–128 (pp. 4–5): The line-intersect method is a widely used approach to quantify FWD lying on the ground in a forest stand (Warren and Olsen, 1964; Van Wagner, 1968; Brown, 1971). It requires measuring the diameter of each piece of wood at its intersection with a transect line which can be considered as a strip of infinitesimal width containing a series of cross-sectional areas (Van Wagner, 1982). The sum of cross-sectional areas divided by the length of the transect line can be converted to volume by multiplying both numerator and denominator by width. FWD load, or weight per unit ground area, is then obtained from Eq. (1) by multiplying the volume by the specific gravity of wood as follows (Van Wagner, 1982):

$$W = \frac{\pi}{2} \times \sum d^2 \times \frac{\pi}{4} \times \frac{G}{L},$$ (1)

where $W$ is the FWD load, $\frac{\pi}{2}$ is a probability factor that allows to sum the cross-sectional areas as circles, $d$ is the piece diameter, $\frac{\pi}{4}$ is the factor required to convert $d^2$ into circular area, $L$ is the length of the transect line, and $G$ is the specific gravity in units of weight per unit volume. Equation (1) assumes that woody pieces are horizontal and does not account for ground slope. To minimize the bias related to tilted pieces that are less likely to be intercepted by the transect line, $W$ can be multiplied by a correction factor equal to the secant of the piece tilt angle relative to horizontal (Brown and Roussopoulos, 1974). Similarly, a correction factor can be calculated from the ground slope angle as follows (Brown, 1974):

$$s = \sqrt{1 + \tan^2\alpha},$$ (2)

where $s$ is the slope correction factor, $\alpha$ is the ground slope angle (degrees). Consequently,

$$W = \frac{\pi^2 \times G \times \sec h \times \sum d^2 \times s}{8 \times L},$$ (3)

where $h$ is the angle between the piece and the horizontal plane (degrees). Measuring diameter on each intersected piece along a transect line can be tedious and time-consuming, especially where small pieces are abundant. In practice, FWD are tallied by diameter size class using a go/no-go sizing gauge, and the number of intercepts over the transect line is reported for each class (Brown, 1974; McRae et al., 1979). Therefore, the term $\sum d^2$ in Eq. (3) is replaced by $\sum_i n_i \times D_i^2$, where $n_i$ is the number of intercepts over the transect line within the diameter size class $i$, and $D_i$ is the representative class diameter (Van Wagner, 1982). The quadratic mean diameter (QMD) is generally used as the appropriate class diameter so that load for any species can be calculated as follows (Van Wagner, 1982; Nalder et al., 1999):

$$W_i = \frac{\pi^2 \times G_i \times \sec h_i \times n_i \times \text{QMD}_i^2 \times s}{8 \times L},$$ (4)

where $W_i$ is the FWD load (t ha$^{-1}$), $G_i$ is the specific gravity (g cm$^{-3}$), $h_i$ is the piece tilt angle (degrees) of the diameter size class $i$, and $\text{QMD}_i$ is the quadratic mean diameter (cm) of the size class $i$ given by:

$$\text{QMD}_i = \sqrt{\frac{d_i^2}{n_i}}.$$ (5)

Brown, J. K: A Planar Intersect Method for Sampling Fuel Volume and Surface Area, For. Sci., 17, 96–102, doi:10.1093/forestscience/17.1.96, 1971.

Brown, J. K.: Handbook for inventorying downed woody material, U.S. Dept. of Agriculture, Forest Service, Intermountain Forest and Range Experiment Station, Ogden, Utah, Gen. Tech. Rep. INT-16, 24 pp., 1974.

Brown, J. K. and Roussopoulos, P. J.: Eliminating biases in the planar intersect method for estimating volumes of small fuels, For. Sci., 20, 350–356, 1974.

McRae, D. J., Alexander, M. E., and Stocks, B. J.: Measurement and description of fuels and fire behavior on prescribed burns: A handbook, Canadian Forestry Service, Great Lakes Forest Research Centre, Sault Ste. Marie, Ontario, Information Report O-X-287, 44 pp., 1979.

Nalder, I. A., Wein, R. W., Alexander, M. E., and de Groot, W. J.: Physical properties of dead and downed round-wood fuels in the Boreal forests of western and Northern Canada, Int. J. Wildl. Fire, 9, 85–99, doi:10.1071/WF00008, 1999.

Van Wagner, C. E.: The Line Intersect Method in Forest Fuel Sampling, For. Sci., 14, 20–26, 605 doi:10.1093/forestscience/14.1.20, 1968.

Van Wagner, C. E.: Practical aspects of the line intersect method, Canadian Forestry Service, Maritimes Forest Research Centre, Fredericton, New Brunswick, Information Report PI-X-12E, 11 pp., 1982.

Warren, W. G. and Olsen, P. F.: A Line Intersect Technique for Assessing Logging Waste, For. Sci., 10, 267–276, doi:10.1093/forestscience/10.3.267, 1964.

L130. Volume of a sample was first dried and then estimated. Would it underestimate volume of the sample due to shrinkage during drying?

1E. We followed the standard procedure (ASTM International, 2014). Yes, the sample may slightly shrink because of drying. We are interested in the volume of the dry matter (thus without moisture influences) and this procedure is consistently applied and reported within and across studies (e.g., Nalder et al., 1999).

ASTM International: ASTM D2395-14: Standard test methods for density and specific gravity (relative density) of wood and wood-based materials, ASTM International, West Conshohocken, Pennsylvania, United States, doi:10.1520/D2395-14, 2014.

Nalder, I. A., Wein, R. W., Alexander, M. E., and de Groot, W. J.: Physical properties of dead and downed round-wood fuels in the Boreal forests of western and Northern Canada, Int. J. Wildl. Fire, 9, 85–99, doi:10.1071/WF00008, 1999.

L 305. Figure 3.

I would recommend to add data points in the figures. Ranges of the independent variable

1F. We modified Figure 3 in the revision as suggested.

L353 (p. 15):

[Figure]

**in situ biomass measurements**

- ● Magadan site
- ◆ Ust-Yansky site

**Allometric relationships**

- Magadan site ($n$ = 43, DBH: 3.9–52.8 cm)
- Ust-Yansky site ($n$ = 20, DBH: 1.8–18.9 cm)
- Alexander et al. (2012), CK ($n$ = 32, DBH: 0.08–29.3 cm)
- Kajimoto et al. (2006), CK ($n$ = 7, max. DBH: 18.6 cm)
- Kajimoto et al. (2006), OM ($n$ = 6, max. DBH: 16.1 cm)

**Figure 4. Allometry models developed in northeastern Siberia for *Larix cajanderi* relating diameter at breast height (DBH) to (a) stem, (b) branches, (c) foliage, and (d) aboveground tree biomass.** For each model, the number of trees harvested ($n$) and the range or maximum (max.) value of DBH are provided. Regression coefficients are given in Table 5 for allometric equations derived at Magadan and Ust-Yansky sites. For allometric relationships developed in Chersky (CK) and Oymyakon (OM) areas, regression coefficients can be found in Kajimoto et al. (2006) and Alexander et al. (2012).

**Responses to reviewer 2**

The authors of Allometric equations and wood density parameters for estimating aboveground and woody debris biomass in Cajander larch (Larix cajanderi) forests of Northeast Siberia observed mean squared diameter and specific gravity. They developed allometric equations at 25 sites in the Republic of Sakha in Russia. They then make comparisons to allometric equations developed in other studies. Overall the methods appear rigorous. This study provides valuable information from an important yet remote region of the world from which little in-situ data is available and makes the case that further data collection efforts are needed. My major concern is that the claim that the allometric equations presented in the text are more generally applicable needs to be better supported or more nuanced. The equations and the comparison of them to other previously developed equations are not validated against in situ data or across a larger region of space. It seems that the equations from earlier work would be more applicable than those developed by the authors in some more northeastern regions or specific stand types, especially near where they were developed. This is very problematic given the manuscripts focus on providing generalized equations to improve our ability to estimate above-ground biomass in this region.

2A. We agree with the reviewer that pooling data from sites with different forest structure and stand age may not be desirable. We have therefore removed the site-common allometric equation from the revision. By doing so, a main contribution of our revised paper is significantly increasing (from 3 to 5) the number of allometric equations available for Cajander larch forests. We agree with the reviewer that near the sampling sites of these four locations (two previous studies, two newly developed equations), the recommendation is to use the corresponding site-specific equation. A practical challenge arises when scientists would want to measure tree biomass in areas that are further away from those four sampling locations. In this case, we recommend that researchers could use site-specific equations when site parameters (e.g., stand age, DBH distribution, etc.) closely match with one of the sampling sites. When this is not the case, we recommend that researchers calculate tree biomass from the five different equations which will result in uncertainty range (e.g., mean biomass and standard deviation). These changes in the revision alter some of the main conclusions of the original version of the paper, for example those focused on generalized applicability. We rewrote these sections in order to reflect these changes (see our response 2J below).

L95 introduction: I suggest highlighting some of the other roles that this data could play (previously mentioned around L35) to strengthen the introduction.

2B. Thank you for your comment. We rewrote this section as follows:

L96–99 (pp. 3–4): The allometric equations and wood density parameters presented in this work will be of use to researchers that want to quantify aboveground and woody debris biomass in Cajander larch forests of Northeast Siberia. This data will facilitate improved quantification and understanding of the dynamics of these two major boreal forest carbon pools in Northeast Siberia.

L140: Is there a citation for the two equations above. Also, some brief explanation of why the samples were dipped in paraffin could be useful.

2C. We added the appropriate reference and specified why the samples were dipped in paraffin in the revision.

L156–159 (p. 6): After drying, the water displacement method requires to seal the surface of woody pieces to prevent bias in volume determination resulting from water absorption during immersion. Each oven-dried sample was then covered with a thin impermeable layer by immersion in hot liquid paraffin (solidification point 57–60 °C, 0.90 g cm$^{-3}$ at 20 °C), and the mass of the coated piece was measured again before volume determination.

L163–169 (p. 7): Finally, specific gravity of each sample was determined as follows (ASTM International, 2014):

$$G = \frac{K \times m_0}{V_0},$$  (6)

where $G$ is the specific gravity (g cm$^{-3}$), $K$ is a constant equal to 1 when the mass is in grams and the volume is in cubic centimeters, $m_0$ is the oven-dry mass (g), and $V_0$ is the oven-dry volume (cm$^3$) given by:

$$V_0 = m_{w,disp} - \frac{m_{coated} - m_0}{\rho_{paraffin}},$$  (7)

where $m_{w,disp}$ is the mass of water displaced (g), $m_{coated}$ is the mass of the oven-dried sample after immersion in the paraffin (g), and $\rho_{paraffin}$ is the density of the paraffin wax (g cm$^{-3}$).

L205-210: I suggest including a bit more information about this fitting method and selecting the exponent c, including references to other work that uses this method. It could also be useful to include these residual plots in the appendix.

2D. Thank you for your comment. We have included additional references to studies that use weighted nonlinear regression as a fitting method (Brown et al., 1989; Parresol, 1999; Moore, 2010) in the revision. We explained in more detail how we selected the exponent $c$ by (1) approximation of the conditional variance of biomass (Picard et al., 2012), and (2) visual assessment of the plots of the weighted residuals against the fitted values. These plots were included in the appendix of the revision (Figure B1), along with the table with exponent $c$ values for each biomass component and site (Table B1).

L229–231 (p. 9): Originally developed in forestry in the 1960s (Cunia, 1964; Wharton and Cunia, 1987), the weighted least squares method can be used to treat models which have non-random residuals (Brown et al., 1989; Parresol, 1999; Moore, 2010). This method involves assigning each sample a positive weight $w_j$ …

L241–247 (pp. 9–10): For each site and biomass component, the value of $c$ was determined by (1) approximation of the conditional variance of biomass, and (2) visual assessment of the plots of the weighted residuals against the fitted values (Table B1; Fig. B1). In the first approach, we followed Picard et al. (2012) by dividing DBH values in $K$ classes centered on $DBH_k$, where $k = \{1,…,K\}$. We selected $K = 5$ in this study and visually checked that the power model was appropriate for modeling the residual variance. A linear regression was fitted between the standard deviation of biomass and the median DBH of each class $k$ using a double logarithmic transformation and the value of the exponent $c$ was approximated as the slope of this regression.

Brown, S., Gillespie, A. J. R., and Lugo, A. E.: Biomass estimation methods for tropical forests with applications to forest inventory data, Forest Sci., 35, 881–902, doi:10.1093/forestscience/35.4.881, 1989.
Moore, J. R.: Allometric equations to predict the total above-ground biomass of radiata pine trees, Ann. For. Sci., 67, 806, doi:10.1051/forest/2010042, 2010.
Parresol, B. R.: Assessing tree and stand biomass: a review with examples and critical comparisons, Forest Sci., 45, 573–593, doi:10.1093/forestscience/45.4.573, 1999.
Picard, N., Saint-André, L., and Henry, M.: Manual for building tree volume and biomass allometric equations: from field measurements to prediction, Food and Agricultural Organization of the United Nations, Rome, Italy, and Centre de Coopération Internationale en Recherche Agronomique pour le Développement, Montpellier, France, 215 pp., 2012.

L446 (pp. 21–23):

Appendix B

**Table B1. Values of the exponent $c$ used in the weighted nonlinear regressions of component biomass against diameter at breast height (DBH) in which the weightings were inversely proportional to $DBH^{2c}$.** The number of observations ($n$) is provided for each regression.

| Biomass component | Magadan | | Ust-Yansky | |
|---|---|---|---|---|
| | $n$ trees | $c$ | $n$ trees | $c$ |
| Stem | 43 | 2 | 20 | 2.5 |
| Stem wood | 32 | 2 | 20 | 2.5 |
| Stem bark | 32 | 2.5 | 20 | 2 |
| Branches | 43 | 2 | 20 | 1.5 |
| Foliage | 43 | 1.5 | 20 | 1.5 |
| Aboveground | 43 | 2 | 20 | 2 |

[Figure]

[Figure]

**Figure B1. Weighted residuals against fitted values from the weighted nonlinear regressions relating diameter at breast height (DBH) to a)-(b) stem, (c)-(d) stem wood, (e)-(f) stem bark, (g)-(h) branches, (i)-(j) foliage, and (k)-(l) total aboveground tree biomass in Magadan and Ust-Yansky sites, respectively.** Weightings were inversely proportional to $DBH^{2c}$.

Figure 3: It would be good to include the data points on this plot where possible and the standard error envelopes for the fits. These lines are also somewhat difficult to distinguish when printed in black and white.

[Figure]

2E. We modified Figure 3 as suggested and changed the colors to improve its readability in gray scale in our revision.

L353 (p. 15):

**Figure 4. Allometry models developed in northeastern Siberia for *Larix cajanderi* relating diameter at breast height (DBH) to (a) stem, (b) branches, (c) foliage, and (d) aboveground tree biomass.** For each model, the number of trees harvested (*n*) and the range or maximum (max.) value of DBH are provided. Regression coefficients are given in Table 5 for allometric equations derived at Magadan and Ust-Yansky sites. For allometric relationships developed in Chersky (CK) and Oymyakon (OM) areas, regression coefficients can be found in Kajimoto et al. (2006) and Alexander et al. (2012).

L206: I think table A1 is important and merits inclusion in the text. It could be interesting to see this comparison done differently. For example, calculating fuel loads at one of the study sites using these different parameters and then plotting the values could better illustrate their importance in percentage terms.

2F. Thank you for this suggestion. We calculated woody debris biomass at 47 of 53 forest stands where *Larix cajanderi* FWD were observed using $M$ factors from this study. For each stand, we derived the percentage difference between these estimates and woody debris biomass computed using $M$ factors from other species and boreal regions provided in Nalder et al. (1999). We plotted this in a new figure as follows:

L307 (p. 12):

[Figure]

**Figure 3. Percentage difference in fine woody debris (FWD) biomass estimates in 47 larch forest stands (Larix cajanderi) near Yakutsk using $M$ factors derived for other species and boreal regions.** Differences were calculated from the estimates based on the $M$ values developed in this study, such that a positive percentage difference reflects a lower biomass estimate. Each box ranges from the first quartile (Q1) to the third quartile (Q3), with the median and mean indicated by a white horizontal line and star. The whiskers extend from Q1 and Q3 to the minimum and maximum defined as Q1−1.5×IQR and Q3+1.5×IQR respectively, where IQR is the interquartile range (Q3−Q1). Outliers above the maximum or below the minimum are indicated by crosses.

L293–298 (p. 11): For 19 of the 20 comparisons, $M$ was greater for *Larix cajanderi* than for the other species, with a mean difference of 23 % (range: -2–46 %) (Table 4). Fine woody debris biomass averaged 2.19 t ha$^{-1}$ in the 47 forest stands sampled in the Republic of Sakha when calculated from size-class specific $M$ values derived in similar Cajander larch forest ecosystems (Table A1). Figure 3 reveals that using $M$ factors derived for other boreal tree species and regions resulted in lower FWD loads between on average 9 % (*Picea mariana*, Northwest Territories) and 28 % (*Larix laricina*, Saskatchewan).

L310: The Magadan site has many more samples than the Yakutian site, although the size range of the available samples varies. Given the conclusion that allometry is somewhat region-specific, it could be interesting to see the result of a fit where this imbalance of samples is corrected using weighting.

2G. That is a good suggestion. Based on the comments of reviewer 1 and you, we have decided to remove the site-common allometry from the revision, and now focus on providing two newly developed allometric equations and their potential use.

Figure 4: For this box plot of the site observations, it would be good to explain the quantiles, etc. shown in the figure caption

2H. Thank you for this suggestion. The statistics displayed in the boxplots are explained in the caption of Figure 3 in the revision:

L310–313 (p. 12): Each box ranges from the first quartile (Q1) to the third quartile (Q3), with the median and mean indicated by a white horizontal line and star. The whiskers extend from Q1 and Q3 to the minimum and maximum defined as Q1−1.5×IQR and Q3+1.5×IQR respectively, where IQR is the interquartile range (Q3−Q1). Outliers above the maximum or below the minimum are indicated by crosses.

L315: Some additional summary information for these 53 sites could be useful (i.e., the mean, sd, and range of dbh)

2I. We agree that this information could be useful. It has been included in a new table (Table 1) that summarize main characteristics of the 53 forest stands where we estimated FWD and tree biomass using the parameters and equations developed in our study.

**Table 1. Stand characteristics of the study sites located in the Republic of Sakha, Russia.** $G$: wood specific gravity; MSD: mean squared diameter; DBH: diameter at breast height; std: standard deviation.

| Forest stands characteristics | Site location | |
|---|---|---|
| | Batamay | Yert |
| Latitude | 63.52° N | 62.02° N |
| Longitude | 129.42° E | 125.79° E |
| Number of stands | 30 | 23 |
| *Number of stands with measurements/estimates for* | | |
| Fine woody debris, *G* and MSD | 4 | 21 |
| Fine woody debris biomass | 25 | 22 |
| DBH, tree biomass | 30 | 23 |
| *Larch trees* | | |
| Mean density ± std (range) (thousand trees ha$^{-1}$) | 10.55 ± 9.93 (1.00–45.33) | 13.50 ± 13.69 (0.17–46.33) |
| Mean tree age ± std (range) (years) | 83.2 ± 40.9 (9–214) | 97.1 ± 39.1 (40–162) |
| Mean DBH ± std (range) (cm) | 5.48 ± 3.56 (1.20–15.08) | 5.09 ± 5.14 (0.22–18.56) |

L315-320, 350-365: I suggest heavily revising these sections of the paper. The claim that the other allometric equations underestimate aboveground biomass or are more generally applicable seems too strong. The actual aboveground biomass of the 53 comparison sites is not truly known. From the text or maps, it's unclear where exactly these test sites are located and how close they are to the sites from Siewert 2015. The comparison to Siewet 2015 is not emphasized in the text. This point would be much stronger if some additional non-allometry-derived data sources, more information from the literature, etc., were included as validation. It could also be interesting to see this comparison done across multiple sites. I imagine these other two equations will perform better in certain areas or stand types. Such a comparison would add more depth to the point about how generalizable each set of equations is.

2J. Thank you for this comment. We agree with you. As explained in our response 2A, we decided to remove the site-common allometry, and focus on our two newly developed allometric equations and their potential use. This changes some of the main conclusions of our work and we have rewritten these sections accordingly.

The aim of comparing the different available equations (3 existing, 2 newly developed) was to show the differences in biomass estimates that result from using the currently available allometric relationships for Eastern Siberian larch forests. We are aware that the estimates at our sampling plots

do not represent ground truth. Yet, we believe that this comparison is of interest as it demonstrates that using the available site-specific allometric equations for Cajander larch forests may lead to significantly different biomass estimates. We already stated that further field efforts to advance our understanding of forest structure and biomass in Cajander larch is necessary and have further reinforced this point in the revision, yet we believe that our paper provides advances in this direction by providing new measurements (for fine woody debris) and increasing the number of available allometric equations.

Following your suggestion, we included additional comparisons with studies that reported aboveground biomass estimates for larch-dominated forest stands in nearby areas. We also included a new map figure (Figure D1) to show the locations of these study sites in reference to our 53 sampling locations.

[revised manuscript text omitted]

L330-340: Interpreting the fitted allometric parameters (i.e., as in Niklas 1994) here and further discussing the differences in climate and other properties between the sites could strengthen the conclusions in this section.

2K. This is a good point. We emphasized differences in stand structure (e.g., tree density, stand age, DBH) and permafrost characteristics that can explain differences between site-specific allometric relationships.

L390–406 (pp. 17–18): Stands structure and dynamics in the Chersky and Oymyakon areas differ with our study sites in the Magadan Oblast and the Ust-Yansky district (Table 2). Allometric equations provided in Alexander et al. (2012) were derived from trees harvested near Chersky in one young

stand (15 years old) and two intermediate-aged stands (60 years old), as well as from a mature stand (155 years old) described in Kajimoto et al. (2006). The mature stand (140 years-old) sampled near Oymyakon was located near the altitudinal treeline of *L. cajanderi* on the middle part of a north-facing slope (10°) and consisted of relatively small trees (mean DBH = 5.93 cm, mean $H$ = 4.31 cm) (Kajimoto et al., 2006). All sites described in this study are underlain by continuous permafrost, however differences in active layer dynamics may contribute to differences in allocation patterns. Soil water and nitrogen (N) availability increase as the active layer deepens, which may allow trees to store more aboveground carbon. In contrast, a shift in proportion of carbon investment from the growth of stem and/or branches to roots development has been reported in northeastern Siberian larch forests growing on permafrost soils with thin active layers (Kajimoto et al., 1999; Kajimoto et al., 2006). This root-oriented biomass allocation pattern is primarily explained as a response of larch trees to N-poor soils (Kajimoto et al., 2010; Osawa et al., 2010). The particular soil conditions that prevail on north-facing slopes of mountainous terrain, such as lower soil temperature and shallower active layers, as well as the limited direct solar radiation may reduce the rate of photosynthesis and respiration of larch trees (Koike et al., 2010). These limited resources of nutrients and light may explain why larch trees at Oymyakon sites (OM) tended to allocate less aboveground biomass per unit DBH (Fig. 4).

Alexander, H. D., Mack, M. C., Goetz, S., Loranty, M. M., Beck, P. S. A., Earl, K., Zimov, S., Davydov, S., and Thompson, C. C.: Carbon Accumulation Patterns During Post-Fire Succession in Cajander Larch (Larix cajanderi) Forests of Siberia, Ecosystems, 15, 1065–1082, doi:10.1007/s10021-012-9567-6, 2012.

Kajimoto, T., Matsuura, Y., Sofronov, M. A., Volokitina, A. V., Mori, S., Osawa, A., and Abaimov, A. P.: Above- and belowground biomass and net primary productivity of a Larix gmelinii stand near Tura, central Siberia, Tree Physiol., 19, 815–822, doi:10.1093/treephys/19.12.815, 1999.

Kajimoto, T., Matsuura, Y., Osawa, A., Abaimov, A. P., Zyryanova, O. A., Isaev, A. P., Yefremov, D. P., Mori, S., and Koike, T.: Size-mass allometry and biomass allocation of two larch species growing on the continuous permafrost region in Siberia, For. Ecol. Manage., 222, 314–325, doi:10.1016/j.foreco.2005.10.031, 2006.

Kajimoto, T., Osawa, A., Usoltsev, V. A., and Abaimov, A. P.: Biomass and Productivity of Siberian Larch Forest Ecosystems, in: Permafrost Ecosystems: Siberian Larch Forests (Ecological Studies), edited by: Osawa, A., Zyryanova, O., Matsuura, Y., Kajimoto, T., and Wein, R., Springer, Dordrecht, The Netherlands, 99–122, doi:10.1007/978-1-4020-9693-8, 2010.

Koike, T., Mori, S., Zyryanova, O. A., Kajimoto, T., Matsuura, Y., and Abaimov, A. P.: Photosynthetic characteristics of trees and shrubs growing on the north- and south-facing slopes in Central Siberia, in: Permafrost Ecosystems: Siberian Larch Forests (Ecological Studies), edited by: Osawa, A., Zyryanova, O., Matsuura, Y., Kajimoto, T., and Wein, R., Springer, Dordrecht, The Netherlands, 273–287, doi:10.1007/978-1-4020-9693-8, 2010.

Osawa, A., Matsuura, Y., and Kajimoto, T.: Characteristics of permafrost forests in Siberia and potential responses to warming climate, in: Permafrost Ecosystems: Siberian Larch Forests (Ecological Studies), edited by: Osawa, A., Zyryanova, O., Matsuura, Y., Kajimoto, T., and Wein, R., Springer, Dordrecht, The Netherlands, 459–481, doi:10.1007/978-1-4020-9693-8, 2010.

**Responses to reviewer 3**

This manuscript describes the size-class-dependent characteristics of fine woody debris according to their detailed field experiment, and the new allometric relationship by using literature data, for the Cajander larch forest in Northeast Siberia. Since such data in this region (central Yakutia) is limited, this manuscript will contribute a lot to future modeling and remote sensing studies in this region. I've already seen the comments from the two reviewers and the author's replies, so I'd like to add some minor comments about what I'm still unsure about.

About fine woody debris: The authors show the MSD and specific gravity of Cajander larch for each diameter size class.

On the other hand, they compared their single factor $M$ with other species in different regions in Table A1, but it is shown in the percentage difference by size class, not the actual values of $M$. I think the single factor M and the fuel load W can be the important outcomes of this study, so I suggest the authors show these results.

3A. Thank you for this suggestion. We have included Table A1 in the main text and modified it to show the actual values of $M$ (see Table 4 below). We have also included a new figure (Figure 3) that shows the percentage difference in FWD biomass in 47 larch forest stands using $M$ factors derived for other species and regions. Differences were calculated from the estimates based on the $M$ values developed in this study (i.e., *Larix cajanderi* in Northeast Siberia). Finally, we added a new table in appendix (see Table A1 below) with biomass means and ranges per diameter size class for each location and species.

L301 (p. 12):

**Table 4. Values of the multiplication factor $M$ from this study and those from other tree species in the Canadian Northwest Territories and Saskatchewan by diameter size class.** $M$ values for *Larix laricina* (tamarack), *Picea glauca* (white spruce), and *Picea mariana* (black spruce) were derived from Eq. (8) using specific gravity ($G$) and mean squared diameter (MSD) values from Nalder et al. (1999). To facilitate comparisons with $M$ values from our study, we used a tilt correction factor (sec h) of 1.13 as suggested by Brown (1974).

| Location | Species | Value of $M$ per diameter size class (II–V) | | | |
|---|---|---|---|---|---|
| | | II | III | IV | V |
| Northeast Siberia | *Larix cajanderi* | 0.628 | 2.89 | 12.1 | 32.3 |
| Northwest Territories | *Picea glauca* | 0.389 | 2.45 | 10.6 | 22.9 |
| | *Picea mariana* | 0.424 | 2.94 | 11.5 | 25.2 |
| Saskatchewan | *Larix laricina* | 0.338 | 1.85 | 9.63 | 31.1 |
| | *Picea glauca* | 0.397 | 2.47 | 10.1 | 19.5 |
| | *Picea mariana* | 0.380 | 2.48 | 10.7 | 23.1 |

L439 (p. 20):

Appendix A

**Table A1. Estimates of fine woody debris (FWD) biomass per diameter size class in 47 larch forest stands (*Larix cajanderi*) near Yakutsk (Republic of Sakha, Russia) calculated using $M$ factors from Table 4.** Estimates are provided as mean ± standard deviation. For each diameter size class (II–V), the average number of pieces tallied along the transect line in the 47 stands is shown, as well as the range in brackets.

| Location | Species | FWD biomass (t ha$^{-1}$) | | | | |
|---|---|---|---|---|---|---|
| | | II | III | IV | V | all classes |
| | | 6.7 (0–36) | 9.7 (0–44) | 1.0 (0–9) | 0.7 (0–4) | 18.0 (1–82) |
| Northeast Siberia | *Larix cajanderi* | 0.14 ± 0.18 | 0.93 ± 1.07 | 0.41 ± 0.71 | 0.71 ± 1.17 | 2.19 ± 2.07 |
| Northwest Territories | *Picea glauca* | 0.09 ± 0.11 | 0.79 ± 0.90 | 0.36 ± 0.62 | 0.50 ± 0.83 | 1.74 ± 1.63 |
| | *Picea mariana* | 0.09 ± 0.12 | 0.95 ± 1.09 | 0.39 ± 0.68 | 0.55 ± 0.91 | 1.99 ± 1.86 |
| Saskatchewan | *Larix laricina* | 0.08 ± 0.10 | 0.60 ± 0.68 | 0.33 ± 0.57 | 0.68 ± 1.13 | 1.68 ± 1.67 |
| | *Picea glauca* | 0.09 ± 0.12 | 0.80 ± 0.91 | 0.35 ± 0.60 | 0.43 ± 0.71 | 1.66 ± 1.55 |
| | *Picea mariana* | 0.09 ± 0.11 | 0.80 ± 0.92 | 0.37 ± 0.63 | 0.51 ± 0.84 | 1.76 ± 1.65 |

L307 (p. 12):

[Figure]

**Figure 3. Percentage difference in fine woody debris (FWD) biomass estimates in 47 larch forest stands (Larix cajanderi) near Yakutsk using $M$ factors derived for other species and boreal regions.** Differences were calculated from the estimates based on the $M$ values developed in this study, such that a positive percentage difference reflects a lower biomass estimate. Each box ranges from the first quartile (Q1) to the third quartile (Q3), with the median and mean indicated by a white horizontal line and star. The whiskers extend from Q1 and Q3 to the minimum and maximum defined as Q1−1.5×IQR and Q3+1.5×IQR respectively, where IQR is the interquartile range (Q3−Q1). Outliers above the maximum or below the minimum are indicated by crosses.

Equation (1):

- Even though the final answer is correct, I strongly suggest you adopt the consistent units in the equation. Specifically, the unit of QMD should be [m], not [cm], and the equation should be multiplied by $10^4$ to convert the unit from [Mg m$^{-2}$] to [Mg ha$^{-1}$]. This will avoid confusion by the readers and avoid careless mistakes in calculation.

3B. Thank you for your comment. We have expressed the specific gravity ($G$) in [g cm$^{-3}$] to be consistent throughout our revised manuscript. However, we think that QMD still need to be expressed in [cm] in the revision. Using both [m] and [cm] in Equation (1) might be confusing for the readers, yet this equation has originally been developed with the diameter of the woody piece ($d$) and the length of the transect line ($L$) expressed in [cm] and [m] (Van Wagner, 1968; 1982). Particularly, the quantity $\frac{\pi^2}{8}$ is a constant, usually referred to as $k$ (Van Wagner, 1982), that allows to retrieve FWD load [t ha$^{-1}$] from $d$ [cm] and $L$ [m]. This equation is consistently applied and

reported within and across studies (e.g., Delisle and Woodard, 1988; Nalder et al., 1999; Alexander et al., 2004; Santín et al., 2015).

$$\frac{[\text{g cm}^{-3}] \times [\text{cm}]^2}{[\text{m}]} \Leftrightarrow \frac{[\text{g}] \times [\text{cm}]^{-1}}{100 \times [\text{cm}]} \Leftrightarrow \frac{[\text{g cm}^{-2}]}{100} \Leftrightarrow \frac{100 \times [\text{t ha}^{-1}]}{100} \Leftrightarrow [\text{t ha}^{-1}]$$

Alexander, M. E., Stefner, C. N., Mason, J. A., Stocks, B. J., Hartley, G. R., Maffey, M. E., Wotton, B. M.; Taylor, S. W., Lavoie, N., Dalrymple, G. N.: Characterizing the jack pine – black spruce fuel complex of the International Crown Fire Modelling Experiment (ICFME), Canadian Forestry Service, Northern Forestry Centre, Edmonton, Alberta, Inf. Rep. NOR-X-393, 2004.

Delisle G. P. and Woodard P. M.: Constants for calculating fuel loads in Alberta, Canadian Forestry Service, Northern Forestry Centre, Edmonton, Alberta, Forest Management Note No. 45, 1988.

Nalder, I. A., Wein, R. W., Alexander, M. E., and de Groot, W. J.: Physical properties of dead and downed round-wood fuels in the Boreal forests of western and Northern Canada, Int. J. Wildl. Fire, 9, 85–99, doi:10.1071/WF00008, 1999.

Santín, C., Doerr, S. H., Preston, C. M., and González-Rodríguez, G.: Pyrogenic organic matter production from wildfires: a missing sink in the global carbon cycle, Glob. Change Biol., 21, 1621-1633, doi:10.1111/gcb.12800, 2015.

Van Wagner, C. E.: The Line Intersect Method in Forest Fuel Sampling, For. Sci., 14, 20–26, 605 doi:10.1093/forestscience/14.1.20, 1968.

Van Wagner, C. E.: Practical aspects of the line intersect method, Canadian Forestry Service, Maritimes Forest Research Centre, Fredericton, New Brunswick, Information Report PI-X-12E, 11 pp., 1982.

- Secant (sec) should be in non-italic.

3C. We changed this in the revision.

L117 (p. 4): $W = \frac{\pi^2 \times G \times \sec h \times \sum d^2 \times s}{8 \times L}$,     (3)

- Is $G_i$ the arithmetic mean of $G$ (specific gravity) within the diameter size class $i$?
  3D. $G_i$ is the specific gravity of the diameter size class $i$ for a given species and location. Our values are shown in Table 3. They were indeed derived by calculating the arithmetic mean of specific gravity within each size class.

- Is $h_i$ the arithmetic mean of $h$ (piece tilt angle) within the diameter size class $i$? If yes, is it mathematically correct to calculate the secant using the arithmetic mean value of h for obtaining the fuel load?
    - For example, If $h$ takes 0 degrees and 180 degrees, the arithmetic mean of them can be 90 degrees.
    - Besides, according to Fig. 2, $h$ is always related to the diameter of each sampled piece, so I think the product of the diameter and sec $h$ should be used for the statistical calculation.

3E. Thank you for your comment. The basic equation developed by Van Wagner (1968) to retrieve woody debris load using the line-intersect approach assumed that sampled pieces lie horizontally on the ground. If pieces are tilted, due to ground slope or because the piece is partially hanging, they are less likely to be intercepted by the transect line. h is the angle between the piece and the horizontal plane and is therefore not related to the diameter of the piece (h < 90°). To minimize the bias related to non-horizontal pieces, $W$ can be multiplied by a correction factor equal to the secant of the angle of tilt from horizontal (h) (Brown, 1974). Brown and Roussopoulos (1974) showed that the average correction factor for naturally fallen branches in American conifer forests ranged between 1.09 (h ≈ 23°) and 1.21 (h ≈ 34°). Tilted bias can be larger in fresh logging slash (correction factor as high as 1.38) where smaller pieces are attached to larger ones. $h_i$ is the specific tilt angle for the diameter size class $i$. It can be derived from the arithmetic mean of h within the size class $i$ (e.g., Nalder et al., 1999).

Brown, J. K.: Handbook for inventorying downed woody material, U.S. Dept. of Agriculture, Forest Service, Intermountain Forest and Range Experiment Station, Ogden, Utah, Gen. Tech. Rep. INT-16, 24 pp., 1974.

Brown, J. K. and Roussopoulos, P. J.: Eliminating biases in the planar intersect method for estimating volumes of small fuels, For. Sci., 20, 350–356, 1974.

Van Wagner, C. E.: The Line Intersect Method in Forest Fuel Sampling, For. Sci., 14, 20–26, 605 doi:10.1093/forestscience/14.1.20, 1968.

Nalder, I. A., Wein, R. W., Alexander, M. E., and de Groot, W. J.: Physical properties of dead and downed round-wood fuels in the Boreal forests of western and Northern Canada, Int. J. Wildl. Fire, 9, 85–99, doi:10.1071/WF00008, 1999.

- If $N$ represents the (total) number of intercepts over the length of the transect line, what does $N_i$ mean?

3F. $N_i$ represents the number of intercepts per diameter size class $i$ over the length of the sample line. We clarified this in the revision.

L121–123 (p. 4): Therefore, the term $\sum d^2$ in Eq. (3) is replaced by $\sum_i n_i \times D_i^2$, where $n_i$ is the number of intercepts over the transect line within the diameter size class $i$, and $D_i$ is the representative class diameter (Van Wagner, 1982).

Equation (2):

- I suggest using a single character (e.g., α) instead of "*slope*" to represent the ground slope.

3G. Thank you for this suggestion. We changed this in the revision.

L115–116 (p. 4): $s = \sqrt{1 + \tan^2\alpha},$     (2)

where $s$ is the slope correction factor, $\alpha$ is the ground slope angle (degrees).

- $(\tan \alpha)^2$ is generally written as $\tan^2\alpha$.

3H. We rewrote Equation (2) as suggested.

L115–116 (p. 4): $s = \sqrt{1 + \tan^2\alpha},$     (2)

where $s$ is the slope correction factor, $\alpha$ is the ground slope angle (degrees).

Equation (3) and L140:

- The authors use two characters to represent the specific gravity. One is $G$ in equation (1) and L104 (kg m$^{-3}$), and another is $S$ here (g cm$^{-3}$).

3I. Thank you for this comment. We have used a single character to represent the specific gravity in the revision (i.e., $G$ in g cm$^{-3}$).

L164–165 (p. 7): $G = \frac{K \times m_0}{V_0}$, (6)

where $G$ is the specific gravity (g cm$^{-3}$), …

L158-159, equation (5):

▪ Does a single factor *M* represent the fuel loads per intercept (sample) on the transect line? Please explain this concept concisely since the reference (Nalder et al., 1999) was not accessible from my environment.

3J. $M$ is a multiplication factor introduced by Nalder et al. (1999) to simplify woody debris loads calculations. It combines the values of specific gravity, tilt angle, and MSD, as shown in Equation (5). For each diameter size class $i$, woody debris loads are then obtained as follows:

$W_i = \frac{n_i \times M_i}{L}$,

where $W_i$ is the woody debris load (t ha$^{-1}$) for the diameter size class $i$, $n_i$ is the number of intercepts within the size class $i$, $M_i$ is the appropriate multiplication factor (g cm$^{-1}$) derived from $G_i$ (g cm$^{-3}$), $h_i$ (degrees), MSD$_i$ (cm$^2$), and $L$ is the length of the transect line (m). It represents the woody debris load per intercept per meter of transect.

L184–196 (pp. 7–8): To simplify FWD loads calculations, size-class specific values of specific gravity, MSD, and tilt angle can be combined into a single factor $M$ (g cm$^{-1}$) as follows (Nalder et al., 1999):

$M_i = \frac{\pi^2 \times G_i \times \sec h_i \times \text{MSD}_i}{8}$. (8)

While tilt bias can be significant in fresh logging slash where smaller pieces are attached to larger ones (e.g., correction factor as high as 1.38) (Brown and Roussopoulos, 1974), a FWD inventory made in harvested and natural Canadian boreal forest sites indicated that it was generally not large (Nalder et al., 1999). To facilitate comparisons with other species and regions, we used a tilt correction factor of 1.13 as suggested by Brown (1974). The single factor $M$ defined in Eq. (8) represents the load per intercept per meter of transect and can then be used to calculate FWD loads for each diameter size class $i$ using the formula:

$W_i = \frac{n_i \times M_i}{L}$. (9)

We used Eq. (9) to calculate FWD loads at 47 of 53 forest stands where *Larix cajanderi* fine woody debris were observed (Table 1) using $M$ factors derived from this study. For each site, we derived the percentage difference between these estimates and FWD loads computed using $M$ factors from other species and boreal regions.

Nalder, I. A., Wein, R. W., Alexander, M. E., and de Groot, W. J.: Physical properties of dead and downed round-wood fuels in the Boreal forests of western and Northern Canada, Int. J. Wildl. Fire, 9, 85–99, doi:10.1071/WF00008, 1999.

- If you share the same units with equation (1), $G_i$ has the unit of [Mg m$^{-3}$], and MSD$_i$ might have the unit of [cm$^2$]. However, the author specified that $M$ has the unit of [g cm$^{-1}$]. In this case, the units of the left and right sides of equation (5) are inconsistent. I suppose the unit of $G_i$ in equation (5) would be [g cm$^{-3}$], or it should be $S_i$ according to equation (3).

3K. Thank you for your comment. Mg m$^{-3}$ is equivalent to g cm$^{-3}$, but we changed the unit of $G_i$ as g cm$^{-3}$ in the revision so that both sides of Equation (5) have similar units.

L125–126 (pp. 4–5): $W = \dfrac{\pi^2 \times G_i \times \sec h_i \times n_i \times \mathrm{QMD}_i{}^2 \times s}{8 \times L}$,       (4)

where $W_i$ is the FWD load (t ha$^{-1}$), $G_i$ is the specific gravity (g cm$^{-3}$), …

- As pointed out in equation (1), I still wonder whether the use of "sec $h_i$" is mathematically correct if $h_i$ represents the arithmetic mean of $h$ in class $i$.

3L. Please see our response 3E. The tilt correction factor (sec h) has consistently been applied and reported within and across studies (e.g., Brown and Roussopoulos, 1974; Nalder et al., 1999).

Brown, J. K. and Roussopoulos, P. J.: Eliminating biases in the planar intersect method for estimating volumes of small fuels, For. Sci., 20, 350–356, 1974.

Nalder, I. A., Wein, R. W., Alexander, M. E., and de Groot, W. J.: Physical properties of dead and downed round-wood fuels in the Boreal forests of western and Northern Canada, Int. J. Wildl. Fire, 9, 85–99, doi:10.1071/WF00008, 1999.

3M. Based on the comments of reviewer 1 and you, we articulated the section *2.1 Fine woody debris sampling* as follows:

L102–128 (pp. 4–5): The line-intersect method is a widely used approach to quantify FWD lying on the ground in a forest stand (Warren and Olsen, 1964; Van Wagner, 1968; Brown, 1971). It requires measuring the diameter of each piece of wood at its intersection with a transect line which can be considered as a strip of infinitesimal width containing a series of cross-sectional areas (Van Wagner, 1982). The sum of cross-sectional areas divided by the length of the transect line can be converted to volume by multiplying both numerator and denominator by width. FWD load, or weight per unit ground area, is then obtained from Eq. (1) by multiplying the volume by the specific gravity of wood as follows (Van Wagner, 1982):

$$W = \frac{\pi}{2} \times \sum d^2 \times \frac{\pi}{4} \times \frac{G}{L}, \tag{1}$$

where $W$ is the FWD load, $\frac{\pi}{2}$ is a probability factor that allows to sum the cross-sectional areas as circles, $d$ is the piece diameter, $\frac{\pi}{4}$ is the factor required to convert $d^2$ into circular area, $L$ is the length of the transect line, and $G$ is the specific gravity in units of weight per unit volume. Equation (1) assumes that woody pieces are horizontal and does not account for ground slope. To minimize the bias related to tilted pieces that are less likely to be intercepted by the transect line, $W$ can be multiplied by a correction factor equal to the secant of the piece tilt angle relative to horizontal (Brown and Roussopoulos, 1974). Similarly, a correction factor can be calculated from the ground slope angle as follows (Brown, 1974):

$$s = \sqrt{1 + \tan^2\alpha}, \tag{2}$$

where $s$ is the slope correction factor, $\alpha$ is the ground slope angle (degrees). Consequently,

$$W = \frac{\pi^2 \times G \times \sec h \times \sum d^2 \times s}{8 \times L}, \tag{3}$$

where $h$ is the angle between the piece and the horizontal plane (degrees). Measuring diameter on each intersected piece along a transect line can be tedious and time-consuming, especially where small pieces are abundant. In practice, FWD are tallied by diameter size class using a go/no-go sizing gauge, and the number of intercepts over the transect line is reported for each class (Brown, 1974; McRae et al., 1979). Therefore, the term $\sum d^2$ in Eq. (3) is replaced by $\sum_i n_i \times D_i^2$, where $n_i$ is the number of intercepts over the transect line within the diameter size class $i$, and $D_i$ is the representative class diameter (Van Wagner, 1982). The quadratic mean diameter (QMD) is generally used as the

appropriate class diameter so that load for any species can be calculated as follows (Van Wagner, 1982; Nalder et al., 1999):

$$W_i = \frac{\pi^2 \times G_i \times \sec h_i \times n_i \times QMD_i^2 \times s}{8 \times L},$$ (4)

where $W_i$ is the FWD load (t ha$^{-1}$), $G_i$ is the specific gravity (g cm$^{-3}$), $h_i$ is the piece tilt angle (degrees) of the diameter size class $i$, and QMD$_i$ is the quadratic mean diameter (cm) of the size class $i$ given by:

$$QMD_i = \sqrt{\frac{d_i^2}{n_i}}.$$ (5)